# Applicability of LC-QToF and Microscopical Tools in Combating the Sophisticated, Economically Motivated Adulteration of Poppy Seeds

**DOI:** 10.3390/foods12071510

**Published:** 2023-04-03

**Authors:** Bharathi Avula, Kumar Katragunta, Sebastian John Adams, Yan-Hong Wang, Amar G. Chittiboyina, Ikhlas A. Khan

**Affiliations:** 1National Center for Natural Products Research, School of Pharmacy, University of Mississippi, University, MS 38677, USA; kkatragu@olemiss.edu (K.K.); jasabest@olemiss.edu (S.J.A.); wangyh@olemiss.edu (Y.-H.W.); amar@olemiss.edu (A.G.C.); 2Division of Pharmacognosy, Department of Biomolecular Sciences, School of Pharmacy, University of Mississippi, University, MS 38677, USA

**Keywords:** opiates, microscopy, LC-QToF, adulteration, quantification, validation, chemical profiling

## Abstract

Morphine and codeine are the two principal opiates found in the opium poppy (*Papaver somniferum* L.) and are therapeutically used for pain management. Poppy seeds with low opiates are primarily used for culinary purposes due to their nutritional and sensory attributes. Intentional adulteration of poppy seeds is common, often combined with immature, less expensive, exhausted, or substituted with morphologically similar seeds, viz., amaranth, quinoa, and sesame. For a safer food supply chain, preventive measures must be implemented to mitigate contamination or adulteration. Moreover, the simultaneous analysis of *P. somniferum* and its adulterants is largely unknown. Pre- and post-processing further complicate the alkaloid content and may pose a significant health hazard. To address these issues, two independent methods were investigated with eight botanically verified and fifteen commercial samples. Microscopical features were established for the authenticity of raw poppy seeds. Morphine, codeine, and thebaine quantities ranged from 0.8–223, 0.2–386, and 0.1–176 mg/kg, respectively, using LC-QToF. In most cases, conventional opiates have a higher content than papaverine and noscapine. The analytical methodology provided a chemical profile of 47 compounds that can be effectively applied to distinguish poppy seeds from their adulterants and may serve as an effective tool to combat ongoing adulteration.

## 1. Introduction

Poppy seeds are a product of the opium poppy (*Papaver somniferum* L.), which belongs to the Papaveraceae family. *P. somniferum* has been cultivated for its seeds and their recreational, analgesic, and narcotic properties since prehistoric times [1,2]. These are tiny, edible, kidney-shaped seeds harvested as a source of alkaloid compounds for the pharmaceutical industry and as a source of poppy seeds for the food industry [3]. The sap, or latex, of the poppy plant, when dried to yield opium, contains multiple alkaloids, including opiates (OAs) such as morphine, thebaine, codeine, papaverine, and narcotine [4,5]. About 218 species belong to the genus *Papaver* [6,7], and 170 alkaloids have been reported from this genus [8]. Among these species, only *Papaver somniferum* L. and *Papaver setigerum* DC. (a synonym of *Papaver somniferum* subsp. *setigerum* (DC.) Arcang.) [9,10,11,12] produce higher quantities of narcotic substances as their secondary metabolites. Hence, the former species has been commonly cultivated. Although the seeds may contain low levels of opiates, contamination of residual poppy latex during harvesting and processing is known to contribute to higher levels of opiates [3,4,5,13].

A wide spectrum of color variants of poppy seeds, from white to yellow–brown, red–brown, red–lilac, blue–grey, and dark blue to black, are found in commerce [14]. The blue or black seeds are primarily used in Europe and the US by sprinkling on baked goods, whereas white (closer to beige) or yellow seeds, which are milder than the black seeds, are commonly used in South Asia, specifically in Indian cuisine, for flavoring and thickening purposes. Finally, the lesser-known brown or maroon seeds come from Turkey. All these seed types come from multiple *P. somniferum* varieties, commonly called the poppy plant, and can easily substitute one for the other. The occurrence data on opium alkaloids show poppy seed samples with high concentrations of morphine, codeine, and thebaine. The opium alkaloid content and profile appear to be related to the country of origin of the poppy seed samples and ultimately, to the poppy variety [4]. Currently, 68 varieties are listed on the EU Plant Variety Catalogue [15]. European Regulation (EU) 2021/2142, published on 3 December 2021 and effective from 1 July 2022, sets maximum levels for OAs expressed in morphine equivalents (morphine + 0.2 codeine) as 1.5 mg/kg for bakery and (or) derived products and 20 mg/kg for whole, ground, or milled poppy seeds [16].

Poppy seeds are not currently controlled in the United States and Canada but are prohibited in many countries, including Korea, Taiwan, Singapore, China, and Saudi Arabia, as they primarily contain morphine, codeine, and other narcotics. On the contrary, the opium poppy is legally grown and widely used in many countries, especially in Central Europe, South Asia, and elsewhere. In 2014, the European Commission recommended good practices to prevent and reduce the presence of opium alkaloids in poppy seeds and their finished products [17]. Various factors, such as cultivar variety, geographical location, time of harvesting, and processing [18], are reported to affect the alkaloid content in opium poppy seeds. Mature poppy seeds obtained from capsules of different varieties of *P. somniferum* and seed oil are directly marketed to consumers for processed foods, such as baked goods, desserts, and other culinary products [4,18]. However, according to the reported studies [4,18,19,20], the content of OAs varied in poppy seed samples sourced from the European market.

The commonly used detection methods are high-performance liquid chromatography with ultraviolet detection (HPLC-UV) [21,22,23], gas chromatography–mass spectrometry (GC–MS) [12,18,24,25,26,27,28,29,30,31,32,33,34], liquid chromatography coupled with tandem mass spectrometry (LC–MS/MS) [19,20,35,36,37,38,39], and LC hyphenated with a time of flight mass spectrometry (LC–QToF-MS) [40,41,42]. GC–MS and LC–MS/MS are more selective and sensitive than HPLC–UV. GC–MS has the drawback of requiring an additional derivatization step to obtain the desired sensitivity. GC–MS, LC–MS/MS, or liquid chromatography combined with high-resolution mass spectrometry (LC–HRMS) techniques are widely applied for the determination of OAs and metabolites in human biological samples such as urine, plasma, serum, hair, saliva, and cerebrospinal fluids [41,42] where positive opiate drug test results have been found [27,28,29,30,31,35,43]. These techniques are also used for metabolite profiling of poppy cultivars or opium latex [12,36,40]. On the other hand, poppy seeds are used in food processing, including bakery products, toppings for dishes, cakes, desserts [23,32], and tea beverages [20,33,44]. Recent industrial uses of these seeds include preparing dairy products (yogurts) and snacks [45]. Schulz et al. used Fourier transform (FT) infrared spectroscopy and NIR-FT-Raman spectroscopy techniques to identify and quantify five important alkaloids (morphine, codeine, thebaine, papaverine, and noscapine) in forty-five samples of poppy capsules, milk, and extracts. Using this method, discrimination between “low-alkaloid” (0.10–0.26 mg/g) and “high-alkaloid” (3.5–10.8 mg/g) plants can be easily achieved [46].

The poppy variety, geographical origin, harvest time, and external contamination can significantly influence the OA content. [19,34,38]. Bjerver et al. determined morphine levels in five poppy seed samples (three blue and two white) and found considerable amounts of morphine, especially in the three blue poppy seeds (9.4–374 μmol/kg) [34]. Lopez et al. (2018) [19] reported results obtained for 41 retail samples from the Netherlands (32 samples), Germany (8 samples), and Italy (one sample). Of these, 32 blue samples, three white poppy seed samples, ground seed samples, one ready-to-use poppy seed filling for the bakery, and two ready-to-eat products were analyzed for six alkaloids. All analyzed samples contained morphine between 0.2–241 mg/kg. Codeine (<0.1–348 mg/kg) and thebaine (<0.1–106 mg/kg) were identified in 78% of the samples. In four samples of black poppy seeds, Hayes et al. (1987) [25] found morphine and codeine in the range of 17–294 mg/kg and 3–14 mg/kg, respectively. The results obtained by Starnska et al. (2013) [39] indicated that the alkaloids morphine, codeine, narcotine, papaverine, and thebaine ranged from 3327–17175, 172–1767, 24–1675, 2–689, and 0–1701 μg/g, respectively, in the selected 15 poppy cultivars.

Sometimes, to increase profits, there is a possibility that the poppy seeds have been adulterated with locally available, less expensive seeds such as amaranth, cumin, chia, sesame, or other seeds that resemble poppy seeds in color and size, thereby increasing the difficulty of determining the authenticity of the poppy seeds, including artificially colored seeds [47,48]. In addition, DNA barcoding methods were successfully implemented to detect adulterants of the *Papaver* genus and helped differentiate *P. somniferum* from other varieties [49,50]. Poppy seed pods are harvested to extract opiate-containing latex before maturity, while culinary poppy seeds are harvested after the seed pods have matured and dried. There have been cases of poppy seed adulteration when immature by-product seeds from pharmaceutical processing with a higher content of opium alkaloids were mixed with food-grade mature poppy seeds. Often, the poppy seeds are heterogenous, having differences in size, color, and taste, and may be contaminated with poppy latex or other impurities [51].

Culinary poppy seeds are frequently mixed with low-quality seeds of cultivars with higher alkaloid content or with exhausted seeds following the extraction of opiates. The mixed product’s decreased quality and increased OA content represent a potential health hazard for consumers [19,52,53]. Even if establishing the detailed external morphological features of botanically verified materials would serve for the proper identification of raw materials, the development of a sensitive analytical method would address the overall quality of the poppy seeds in commerce or in poppy-derived products. Moreover, assessing extraction efficiency with various solvents and removing opiates from post-processed poppy seeds would be beneficial to differentiate naturally latex-contaminated seeds from seeds artificially spiked with opiates. Nevertheless, an untargeted analysis and the development of comprehensive alkaloid profiles could help discriminate between species in seeds of the genus *Papaver* and for detecting any added adulterants or substituents.

## 2. Materials and Methods

### 2.1. Chemicals and Standards

Representative images of poppy seeds and the chemical structures of five OAs are shown in Figure 1 and Figure 2. Morphine, codeine, thebaine, noscapine, papaverine, morphine-d_3,_ and codeine-d_3_ were purchased from Sigma (St. Louis, MO, USA). The reference compounds were more than 99% pure, with identity and purity confirmed by chromatographic and spectral data (HRMS). HPLC-grade solvents such as acetonitrile, methanol, isopropanol, and formic acid were procured from Fisher Scientific, Waltham, MA, USA. Ultrapure water (18.2 MΩ.cm) was purified using the Milli-Q system (Millipore, Bedford, MA, USA).

### 2.2. Preparation of Standard Solutions

Stock solutions of standard compounds were prepared at a concentration of 1 μg/mL in methanol. Calibration curves ranged from 0.5–250 ng/mL at five different concentration levels. All standard solutions were stored in amber vials at 4 °C.

### 2.3. Plant Materials

Seed samples of *P. somniferum* (#2546, 7528, 12750, and 22469) were sourced from Hamdard University (Karachi, Pakistan), CRISM (Delhi, India), and the Missouri Botanical Garden (St. Louis, MO, USA). Samples of poppy seeds (#5552, 5558, 16725, 24790, 24791, 24792, 24793, 25064, 25107, and 25124) and other plant seed samples used as adulterants or substituents: *Amaranthus cruentus* L. (#22982 and 25225), *A. caudatus* L. (#23019 and 25226), *A. paniculatus* L. is a synonym of *A. cruentus* L. (#25223), *Salvia hispanica* L. (#22449 and 25145), and *Nigella sativa* L. (Black cumin) (#4978, 7516) seed samples were obtained from CRISM (Delhi, India), and two samples of *Sesamum indicum* L. (Sesame) (#2519 and 22448) seed samples were collected from Pakistan and Missouri Botanical Garden (St. Louis, MO, USA). *Chenopodium quinoa* Willd. (quinoa) seed samples (#6146 and 22444) were procured from the Missouri Botanical Garden (St. Louis, MO, USA). *Nigella sativa* (#22471) and *Eschscholzia californica* Cham. (golden poppy) seed samples (#24794) were obtained commercially.

Poppy seed samples (#2116–2021, 2123, and 2124) used in food were commercially procured from Amazon in November 2022. These samples were deposited at the NCNPR’s botanical repository, the University of Mississippi, MS, USA.

### 2.4. Plant Sample Preparation

Dry-ground seed samples were weighed separately for about 1000 mg, 100 mg, and 10 mg each. 100 μL of the internal standards (2.5 μg/mL) were added to these samples and sonicated in a 2.5 mL volume of methanol containing 1% formic acid for about 30 min, followed by their being centrifuged for 15 min at 959× *g*. The supernatant solution was transferred to a 10 mL volumetric flask. The procedure was repeated three more times with fresh solvent, and their supernatants were combined in a volumetric flask and adjusted to achieve a final volume of 10 mL. Before injection, approximately 2 mL of the solution was passed through a 0.45 µm PTFE membrane filter. The first 1.0 mL was discarded, and the remaining volume of about 1 mL was collected in an LC sample vial.

### 2.5. Micro Morphology Analysis

The external morphology of seeds was observed under low magnification using a NIKON SMZ-U (Japan) stereomicroscope with photo capture by the camera (NIKON DS-Fi1) attached to the microscope and processed with the software NIS-Element BR. Specimens were fixed in formaldehyde alcohol acetic acid (FAA) for high-level magnification using scanning electron microscopy (SEM) analysis. These samples were passed through a 100% ethanol solution and dehydrated using a 1 h and 15 min programmed critical point dryer (Leica CPD300, Wetzlar, Germany) under CO_2_. Dried samples were placed on double-sided adhesive carbon tape and pasted on the aluminum stubs. For platinum coating, these stubs were placed on a Desk V HP sputter coater (Denton Vacuum, NJ, USA) supplied with argon gas. The prepared samples were used for SEM analysis using a JSM-7200FLV field-emission SEM (JEOL Ltd., Tokyo, Japan).

### 2.6. Instrumentation and Analytical Conditions

#### Liquid Chromatography–Quadrupole Time of Flight Mass Spectrometry (LC–QToF)

The liquid chromatographic system was an Agilent Series 1290 system comprised of a binary pump, a vacuum solvent micro-degasser, a 100-well autosampler, and a thermostatically controlled column compartment. Separation was achieved on an Agilent Poroshell 120 EC-18 (2.1 × 150 mm, 2.7 µm) column at a 0.2 mL/min flow rate. The mobile phase consisted of water with 0.1% formic acid (A) and acetonitrile with 0.1% formic acid (B) with the following gradient elution: 99% A/1% B, isocratic for 3 min, 45% B in the next 27 min, and 100% B in the next 10 min. A 5 min wash followed each run with 100% acetonitrile and an equilibration period of 5 min with 99% A/1% B. One microliter of the sample was injected at a column temperature of 40 °C. The mass spectrometric analysis was performed with a QToF–MS/MS (Model #G6545A, Agilent Technologies, Palo Alto, CA, USA) equipped with an ESI source under the following conditions: drying gas (N_2_) flow, 13.0 L/min; drying gas temperature, 325 °C; nebulizer, 27 psig; sheath gas temperature, 325 °C; sheath gas flow, 11 L/min; capillary, 3500 V; skimmer, 65 V; Oct RF V, 750 V; and fragmentor voltage, 150 V. The sample collision energy was set at 45 eV. All operations, data acquisition, and data analysis were controlled by Agilent MassHunter Acquisition Software Ver. A.10.00, with further data processing by MassHunter Qualitative Analysis Software Ver. B.10.0. Each sample was analyzed in positive and negative modes in the range of *m/z* = 50–1700. Accurate mass measurements were obtained employing ion correction methodologies using mass references at *m/z* 121.0509 (protonated purine) and 922.0098 [protonated hexakis (1H, 1H, 3H-tetrafluoropropoxy) phosphazine or HP-921] in positive ion mode, and mass references at *m/z* 112.9856 (deprotonated trifluoroacetic acid-TFA) and 1033.9881 (TFA adducted HP-921) were used in negative ion mode. The compounds were confirmed in each spectrum.

### 2.7. Method Validation

The LC–QToF method was validated following International Conference on Harmonization (ICH) guidelines regarding precision, accuracy, carryover, stability, and linearity [54]. The limit of detection (LOD) and limit of quantification (LOQ) parameters were determined by injecting a series of diluted solutions with known concentrations. LOD and LOQ were defined as signal-to-noise ratios equal to three and ten, respectively. The accuracy of the assay method was verified in duplicate using two concentration levels of 10 and 100 ng/mL. Intra- and inter-day variation of the assay was determined on three consecutive days with three repetitions each [55].

### 2.8. Post-Processing of Poppy Seeds

#### 2.8.1. Ground vs. Intact Poppy Seeds

To understand the possible role of grinding poppy seeds, 100 mg of intact seeds (#2120PR) and the corresponding ground mixture (#2120PR) were extracted separately with 1% formic acid in methanol (10 mL) in five replicates and analyzed for the content of the five alkaloids: morphine, codeine, thebaine, papaverine, and noscapine.

#### 2.8.2. Washing Effect on Poppy Seeds

To understand the possible impact on OAs content of water washing intact poppy seeds, 500 milligrams of poppy seeds (#2546) were soaked in water (10 mL) and methanol with 1% formic acid (10 mL) separately at different temperatures (cold and hot, 70 °C) and at different time intervals (1, 10, 30, 60, and 120 min; 4, 8, 24, and 48 h; and 4, 6, and 8 days) with constant agitations. At each time increment, 100 µL of an aliquot was collected and analyzed for morphine, codeine, thebaine, papaverine, and noscapine. At the same time, 100 µL of water or methanol (1% formic acid) was added to maintain the constant final volume (10 mL). After 8 days of washing, all samples were dried, ground, and extracted with 10 mL water or methanol (1% formic acid) separately and analyzed. Effects of solvent, temperature, and extraction time factors on poppy seeds were also investigated.

In another experiment, ground poppy seeds (500 mg, #2546) were soaked in 10 mL each of water and methanol (1% formic acid) at different temperatures (cold and hot, 70 °C) and at different time intervals (1, 10, 30, 60, and 120 min; 4, 8, 24, 48 h; 4, 6, and 8 days) with constant agitations. At each time, 10 mL of solution was removed and analyzed for morphine, codeine, thebaine, papaverine, and noscapine. At the same time, 10 mL of water or acidified methanol was added. This process was continued for 8 days, and after 8 days of thorough washing, the samples were dried and ground. Samples were again extracted with water or methanol (1% formic acid) and analyzed separately. In this study, the OAs in each washing cycle were estimated.

## 3. Results and Discussion

### 3.1. Scanning Electron Microscopic Observation of Poppy and Its Adulteration/Substitution Seeds

*P. somniferum* seeds are of a tiny reniform shape and are noticeably blue/black and white-colored. Seeds are, on average, 1.65 mm long and 1.15 mm wide. The different color seeds are morphologically similar, with white seeds being 0.10–0.25 mm smaller than black seeds. Blue/black/white seeds show a rough surface with a pitted honeycomb-like structure, as shown in Figure 3. The surface is rough in blue–black seeds (Figure 3a–f), whereas in white seeds, the texture is less waxy coated (Figure 3g–h’). The hilum is at the center of the concave curve of the seed (Figure 3a–h). Known adulterant seeds can be differentiated via SEM microscopic characters (Figure 3i–p’) using the features highlighted in Table 1. Microscopically, these adulterant seeds have different seed coat (testa) textures from genuine poppy seeds. *Eschscholzia californica* seeds have a rough and microsculpted surface, but they can be differentiated by the absence of the characteristic honeycomb texture observed in poppy seeds (Figure 3m,m’) and also by the cylindrical shape of the seed. *Amaranthus* spp. seeds have smooth and polished dusty surfaces (Figure 3n,n’) and a reticulate hierarchical structure (Figure 3p,p’) using SEM observation. Other less common adulterants, such as *Nigella sativa* (Figure 3i,i’)*, Chenopodium quinoa* (Figure 3j,j’), *Sesamum indicum* (Figure 3k,k’), and *Salvia hispanica* (Figure 3l,l’) have specific features starkly different from poppy seeds.

### 3.2. Extraction from Poppy Seeds

Extraction efficiency for alkaloids was investigated using the following solvents and solvent mixtures covering a broad range of polarity index values: methanol, acidified methanol (0.1% and 1% formic acid), ethanol, water, and solvent mixtures such as acetonitrile–water (1:1, *v/v*), acetonitrile–water (1:1, *v/v*) with 0.1% formic acid, IPA–methanol (1:1, *v/v*), and IPA–ethanol (1:1, *v/v*) with 0.1% formic acid (Figure 4). Poppy seeds (# 2117PR) were homogenized using a grinder. Approximately 100 mg of the seeds were weighed in duplicate, and the solvent was added. A mixture of deuterated internal standards (morphine-d3 and codeine-d3) was added to each solution; tubes were capped, placed into an ultrasonic bath for 30 min, and centrifuged at 4000 rpm for 15 min. Ultrasound-assisted extraction was carried out and optimized for the above solvents, time of extraction, and concentration of constituents in each organic solvent. Among the different solvents, the optimum extraction conditions were realized when 100 mg of poppy seeds were extracted with 10 mL of methanol containing 1% formic acid (see under the Section 2).

### 3.3. Optimization of Chromatographic Conditions

After several trials, optimized chromatographic conditions were achieved with acetonitrile and water with formic acid in different proportions for the mobile phase and column temperatures at ambient and 40 °C. Mass Hunter Workstation software, including Qualitative Analysis (version B.10.00), was used for processing both raw MS and MS/MS data. This includes molecular feature extraction, background subtraction, data filtering, and molecular formula estimation (using an exact mass of all isotopes, their relative abundances, and all detected adducts with their measured isotopes). A blank sample (methanol) was analyzed under identical instrument settings, and background molecular features (MFs) were removed. This method also involved using the [M + H]^+^ ions in the positive ion mode found in the extracted ion chromatogram (EIC).

### 3.4. Validation Procedure

According to ICH guidelines (Table 2), the newly developed LC–QToF method for the five opiates (OAs) was validated for selectivity, sensitivity, the limit of detection (LOD), the limit of quantification (LOQ), stability, precision, accuracy, specificity, and linearity [54].

The specificity of the method was determined by comparing chromatograms of the blank (methanol) with poppy seed samples and spiked blanks (OAs added to methanol and code #2124PR, 25107). A comparison of methanol blanks with samples and spiked blanks demonstrated the specificity and selectivity of the chosen methodology.

The LOD and LOQ were determined by injecting diluted solutions with known concentrations of each standard. LOD and LOQ were defined as signal-to-noise ratios equal to three and ten, respectively. A five-point calibration curve for the five opioid alkaloids showed a linear correlation between concentration and peak area. Calibration data indicated the linearity (*r*^2^ > 0.99) of the detector response. The detection and quantification limits were 10–25 pg/mL and 25–100 pg/mL, respectively. All samples and standard solutions were analyzed in duplicate.

The accuracy of the assay method was evaluated by spiking two products (#2124PR and #25107) in duplicate using concentration levels of 10 and 100 ng/mL. The method’s accuracy was determined for the related compounds by spiking the sample (#2124PR and #25107) with a known amount of the five opiates standard. These samples were spiked with known amounts of the standard compound mixture and were extracted four times under optimized conditions. These sample’s percentage recovery (%RSD) ranged from 90–109% (1.7–4.4%).

The precision of a method is the degree of agreement among individual analytical results when the procedure is applied repeatedly to multiple samples of each product. The intra- and inter-day precisions were estimated by analyzing multiple replicates of two products (#2124PR and #25107). The intra-day precision of the assay was estimated by calculating the relative standard deviation (%RSD) for the analysis of samples in three replicates (n = 3) of each product, and the inter-day precision was determined by the analysis of three replicates of the same product on three consecutive days. The intra-day %RSD for the replicates was between 0.7 and 3.9%, and the %RSD for the day-to-day replicates was between 1.1 and 3.9%.

From the measured standard deviation (SD) and mean values, precision as relative standard deviation (%RSD) is calculated as %RSD = (SD/mean) × 100.

The sample solution (#2124PR and #25107) and standard solutions (10 and 50 ng/mL) were prepared by the proposed method and subjected to a stability study at room temperature for 72 h. The sample solution was analyzed at the initiation of the study period and at three additional time intervals before 72 h. No significant changes were observed.

### 3.5. Analysis of Poppy Seeds

LC–QToF–MS is a powerful technique offering high sensitivity and selectivity for qualitative and quantitative measurements of analytes of interest in a complex mixture. The high-resolution mass detection in the positive ion mode is useful for detecting alkaloids. In the positive ion mode, protonated species [M + H]^+^ at *m/z* 286.1440, 300.1596, 312.1597, 340.1541, and 414.1545 for morphine, codeine, thebaine, papaverine, and noscapine, respectively, were observed. The combination of retention time and exact mass results are helpful for the general identification of the compounds. The calibration curves were prepared for the compounds listed in Table 2. The validation parameters—retention time, molecular formula, and characteristic fragment ions observed for all compounds—are summarized in Table 3 and Table 4. Retention time, accurate mass, and MS/MS analysis are helpful in the identification of compounds.

From each batch of poppy seeds, two different portions (each weighing ∼100 mg) were ground, extracted, and analyzed. Stranska et al. (2013) reported the alkaloids content from 15 different cultivar samples as 3327–17175 µg/g (morphine), 172–1767 µg/g (codeine), 0–1701 µg/g (thebaine), 2–689 µg/g (papaverine), and 24–1675 µg/g (noscapine) [39]. This is while Lopez et al. (2018) reported that the alkaloids’ content varies from 0.2–241 mg/kg (morphine), <0.1–3.48 mg/kg (codeine), and <0.1–106 mg/kg (thebaine) in 41 different commercially purchased samples from the Netherlands, Germany, and Italy [19]. In our study, 23 poppy seed samples (15 black/blue/brown mix and 8 white/brown mix) were analyzed for the content of five OAs. 17 of the 22 samples showed higher morphine content than any of the other four OAs analyzed. In these 17 samples, the morphine content ranged from 0.8 to 223 μg/g. Two samples (#24792 and #2120PR) showed high codeine content, while codeine content ranged from 0.2–386 μg/g across all 22 samples. One sample (#2117PR) showed a high thebaine content, whereas the thebaine content ranged from 0.1 to 176 μg/g across all 22 samples. Higher concentrations of morphine, codeine, and thebaine were detected in black and black–blue mix samples, while these OAs were lower in white and white–brown mix samples. The other two compounds, papaverine, and noscapine, ranged from DUL—28 μg/g to DUL—73 μg/g, respectively. Sample #6485, labeled “*Papaver* species,” contained all five OAs (Table 3) and showed a profile similar to that of *P. somniferum*. The mean levels of OAs are shown in Table 3, Figure 5 and Figure 6. Differences in the cultivar varieties, harvesting conditions, and post-processing procedures (viz., washing) affect the alkaloids content from the previously published reports. Furthermore, compared to previous reports, the developed method can perform quantitative analysis of opiates as well as chemical profiling to differentiate authentic poppy seeds from adulterants. Nevertheless, high OAs variation in these samples suggests that these poppy seeds may not meet the limits set by European authorities.

### 3.6. Post-Processing Studies of Alkaloids

#### 3.6.1. Alkaloid Content in Ground vs. Intact Poppy Seeds

This experiment was aimed at evaluating alkaloid degradation during the processing of food. Unground and ground seeds were extracted separately with methanol (1% formic acid) in five replicates and analyzed for alkaloid content. The influence of grinding on the OA content was studied. Insignificantly lower concentrations of OAs were found in the ground seeds compared to the unground, intact seeds. Around 10–15% depletion of three alkaloids was observed, accounting for the formation of pseudomorphine (C_34_H_36_N_2_O_6_, *m/z* 569.2646 [M + H]^+^) [56,57], morphine-*N*-oxide (C_17_H_19_NO_4_, *m/z* 302.1387 [M + H]^+^) [56,57], codeine-*N*-oxide (C_18_H_21_NO_4_, *m/z* 316.1543 [M + H]^+^) and thebaine *N*-oxide (C_19_H_21_NO_4_, *m/z* 328.1543 [M + H]^+^). The data also suggested that temperature and light have minor effects on the content of the five studied opiates.

Besides the poppy variety, the harvest method has the strongest influence on the morphine content and leads to the greatest variability in the alkaloid concentration. High OAs concentrations can be attributed to insufficient harvesting practices leading to seed contamination with latex, inadequate cleaning and handling, or intentional spiking with pure standard morphine. The non-uniformity of this contamination leads to inhomogeneities observed within the sample replicates. Similar observations were seen by Sproll et al. (2006 and 2007) [4,19,38,56,58]. When the levels of OAs in unground poppy seeds from five replicates within each batch (#2120) were compared, it was found that there was much variation with the levels of morphine, codeine, thebaine, and noscapine ranging from 0.6–7.6 μg/g, 218–424 μg/g, 12.4–68.2 μg/g, and 0.5–1.4 μg/g, respectively, suggesting possible anomalies in post-harvesting. While in the ground seed samples, there was little variation between the five replicates (#2120), indicating the homogeneity of ground samples. These wide variations observed within the same seed sample suggest the presence of OAs may be due to inadequate cleaning of the seeds, with most of the opium alkaloids coming from debris or residual latex on the seed surface.

#### 3.6.2. Removal of Free Alkaloids by Soaking

Only 40–50% of the five alkaloids are removed with cold water and acidified methanol washes. Seed washing at 70 °C significantly reduced OAs content by 70–80% (Figure 7), and extended washing for 48 h resulted in the complete removal of OAs. Similar observations were made by Sproll et al. (2007) [56].

Very low levels of OAs were detected in studied poppy seed samples using the current analytical methodology. This study confirms that poppy seed morphine or other OAs originate predominantly from external contamination. This was further confirmed by several seed-washing experiments that significantly reduced the OA content. Bjerver et al. (1982) [34] showed that 40% of the total morphine could be removed by a single washing with slightly acidified water. Soaking poppy seeds in water for five minutes removed about 46% of their free morphine and 48% of their codeine [59]. Moreover, the results presented in Figure 6 shows that about 40% of the total opioid content in the seed samples could be isolated from the seeds by a single, simple washing procedure. Alkaloid content in poppy seeds found in this study agrees with prior reports [34,56,59].

### 3.7. Identification and Characterization of Chemical Constituents

In this study, positive ESI tandem mass spectra of the [M + H]^+^ ions of alkaloids were obtained using LC–QToF. High-resolution mass resolving power and accurate mass measurement for MS and MS–MS experiments enable rapid dereplication of various alkaloids within the genus *Papaver*. It generates characteristic fragment ions, which help confirm the known compound in a complex mixture. Possible fragment ions are shown in Table 4. Accurate mass measurements were performed to obtain each analyte’s elemental composition. The diverse class of alkaloids, the major components of poppy seeds, have been categorized into seven groups based on their structural motifs and MS/MS fragmentation patterns, including morphinane, benzylisoquinoline, rhoeadine, tetrahydroisoquinoline, promorphinane, protopine, and benzophenanthridine types. The mass spectral fragmentation patterns of five reference compounds were studied, and based on these patterns, other alkaloids were investigated. Figure 8 shows the extracted ion chromatograms (EIC) of reference standards and poppy seeds (#24792) that have five opioids (OAs).

#### 3.7.1. Fragmentation Patterns of Compounds **1**–**6** (Table 4)

Full-scan mass spectra in the positive ion mode of the LC peak at 3.3 min provided a quasimolecular ion at 𝑚/*z* 154.0863, suggesting a molecular formula with [C_8_H_11_NO_2_]^+^ (calculated to be 154.0863). The MS^2^ spectrum of the ion at 𝑚/*z* 154.0863 generated a series of ions at 𝑚/*z* 137.0598, 119.0488, 109.0647, and 91.0543, which agree with the fragmentation pattern of dopamine. The formation of product ions *m/z* 137 and 119 is attributable to the neutral loss of the amine group [M + H-NH_3_]^+^ and water molecule from the terminus of the dopamine molecule (Table 4) [60].

The major fragmentation pathway of protonated phenylalanine (*m/z* 166.0861) starts with the loss of water and CO to form a fragment ion at *m/z* 120.0801. Further loss of ammonia resulted in a fragment ion with *m/z* 103.0545. Leucine and isoleucine (*m/z* 132.1015) produced two fragments by sequential losses of H_2_O and CO (*m/z* 86.0966) and NH_3_ (*m/z* 69.0696) [60].

Protonated adenosine and guanosine dissociated through decompositions of base-protonated [B + H]^+^ ions by the cleavage of the glycosidic bonds to give the protonated bases with a sugar moiety as the neutral fragment. Fragment ions *m/z* 268.1042 and 284.0890 correspond to the protonated molecular ions of adenosine and guanosine. Ions at *m/z* 136.0619 [M + H-132]^+^ and 152.0567 [M + H-132]^+^ are protonated ions of adenine and guanine, respectively. Typical neutral losses correspond to the molecules -CO, -NH_3_, -CH_2_CH_2_, -NHCH_2_, -NHCO, and -NH_2_CN. The glycosidic C–N bond cleavage is characteristic of two nucleosides. The major fragmentation pathways are ring contraction (RC) and retro-Diels-Alder (RDA) [61].

#### 3.7.2. Fragmentation Patterns of Alkaloids (Compounds **7**–**47**) (Table 4)

Morphinane: The fragments are formed by cleavage of the piperidine ring and loss of an amine (CH_2_CHNHCH_3_, [M-57]^+^). Such a fragment is found for all morphine-like compounds and is crucial in further fragmentation pathways [62].

Morphine and codeine exhibit similar fragmentation patterns, and the major product ions obtained were *m/z* 201.0910 and *m/z* 215.1067, respectively, which were derived from the precursor ion [M + H]^+^ by cleavage of the piperidine ring and consecutive losses of ethylene methylamine [CH_2_CHNHCH_3_]^+^ and CO [62]. Consecutive losses of two water molecules in the case of morphine or methanol and water for codeine result in a cyclopentadiene naphthalene product ion with *m/z* 165.0704 and the elemental composition [C_13_H_9_]^+^. Loss of formaldehyde (H_2_CO) in place of H_2_O produces a dicyclopentadiene benzene product ion with *m/z* 153.0704 and the elemental composition [C_12_H_9_]^+^. The major product ions for thebaine are [M + H-31]^+^, [M + H-46]^+,^ and [M + H-61]^+^, which correspond to the loss of methylamine and one and two methyl groups, respectively [63].

Five cleavages are still found, however, corresponding to those in codeine or morphine, giving ions at M-15 (loss of methyl), M-29, M-43, M-57, M-58, M-59, and M-17 (loss of hydroxyl) in neopine [64].

Promorphinane alkaloid: Salutaridine and salutaridinol differ from the morphine type since the 4,5-epoxy bridge is opened and additional oxygen is present at position 7. The difference between the keto and hydroxy groups at the C-7 position revealed through fragments reveals the characterization of salutaridine and salutaridinol. Initially, the fragmentation of salutaridine is characterized by the loss of methylamine [M + H-31]^+^ from the precursor ion *m/z* 328.1540 [M + H]^+^ followed by the subsequential loss of two methanol groups to form *m/z* 265.0857 and 233.0597. In addition, the key fragment loss of -HCHO (*m/z* 298.1430) confirms the presence of the keto group. Whereas, the formation of the *m/z* 298.1455 fragment reveals the loss of a methanol group followed by methylamine loss (*m/z* 267.1020). The absence of the epoxy bridge facilitates the formation of an isoquinoline fragment at *m/z* 192 [62].

Phthalideisoquinoline: The product ion of noscapine at *m/z* 220.0972 indicates the loss of meconine or 6, 7-dimethoxy-3H-1-isobenzofuranone (C_10_H_10_O_4_, 194.0579 Da) from the precursor ion [M + H]^+^, and subsequent loss of azirine (*m/z* 41.0265, C_2_H_3_N) produces a fragment at *m/z* 179.0716. A minor product ion of noscapine at *m/z* 353.1001 [M + H-61]^+^ is also detected, which can be explained by the loss of aziridine (*m/z* 43.0422, (CH_2_)_2_NH) and water [64].

Benzyltetrahydroisoquinoline and aporphine alkaloids were distinguished by characteristic losses of the NHR_1_R_2_ (R_1_ and R_2_ represent the substituent groups of the nitrogen atom) radical and the fragment ions below *m/z* 200. The MS/MS investigation of benzyltetrahydroisoquinoline and aporphine alkaloids results in fragment ions [M-45]^+^, [M + H-31]^+,^ and [M + H-17]^+^ always being observed initially depending on the number of N-methyl groups. These fragments are attributed to a loss of the NH(CH_3_)_2_ (R_1_ = R_2_ = CH_3_), NH_2_CH_3_ (R_1_ = H, R_2_ = CH_3_), and NH_3_ (R_1_ = R_2_ = H) radicals, respectively. These characteristic ions play a diagnostic role in the discrimination of other alkaloid types. However, the mass spectrum fragmentation behaviors of magnoflorine and corydaline were different from those of the benzyltetrahydroisoquinoline alkaloid because of the conjugate structure. No characteristic ions exist below *m/z* 200, and the product ions are formed mainly by the loss of some substituent groups [65].

Protopine alkaloids (41–44): Product ions were generated by dehydration (*m/z* 336), retro-Diels-Alder (RDA) fragmentation (*m/z* 149 and 206), α-cleavage forming small fragment molecules, and subsequent losses of H_2_O (*m/z* 188) and OH (*m/z* 189), according to Shim et al. 2013 [66], Sai et al., 2020 [67], and Schmidt et al. 2007 [68]. Fragment ions at *m/z* 206.0823 and 149.0603 in the MS/MS spectrum are generated by RDA C ring opening, but given the presence of hydroxyl groups, the product ions at *m/z* 336.1209 and *m/z* 188.0721 are probably formed by loss of H_2_O from the molecular ion and the *m/z* 206.0823 ion.

Sanguinarine (benzophenanthridine-type) was also characterized by the fragmentation of their substituents, with the most abundant ions being formed by the loss of CH_3_ (*m/z* 317.0685), CH_2_O (*m/z* 302.0811), and CO (*m/z* 276). Benzophenanthridine alkaloids contain a large π conjugate system, making the parent nucleus difficult to fragment. The benzophenanthridine alkaloid sanguinarine, containing a methylenedioxy group, would lose carbon monoxide to form a stable ternary oxygen ring [67].

The alkaloids of the rhoeadine type represent a new group of natural bases isolated from plants of the Papaveraceae. These cyclic acetal compounds represent the molecular ion at [M-15]^+^ and the base fragment ion at *m/z* 177. This shows the stable fragment cleavage at C-1 and C-2 positions (C- and D-rings) [69].

Stilbenes: The product ions of narceine from the precursor ions at *m/z* 428.1707 [M + H-18]^+^, *m/z* 383 [M + H-63]^+^, and *m/z* 365 [M + H-81]^+^ can be rationalized by the losses of dimethylamine and one or two water molecules, respectively [19].

Benzylisoquinolines: These correspond to ion types a, [M + H-NH_3_]^+^, and ion types b, c, the protonated parental molecule [M + H]^+^, and ion type d, respectively. The resulting ion at *m/z* 123.04390 constitutes the signature fragment obtained for 1-benzylisoquinolines containing hydroxyl groups at C3′ and C4′. Increases in 14 or 28Da (*m/z* 137 or *m/z* 151) for the equivalent ions indicate single- or double-methylation at C3′ and C4′ [70].

Papaverine is a non-narcotic alkaloid found to be endemic and selectively localized in the latex of the opium poppy. Papaverine readily produced an *m/z* 340.1543 [M + H]^+^ protonated pseudomolecular ion. The fragmentation of the precursor ion can be explained by the loss of a methyl group, resulting in a product ion at *m/z* 324.1236, and the loss of the dimethoxy benzyl moiety to yield the prominent isoquinoline moiety at *m/z* 202.0868 via the C–C bond between the dimethoxy benzyl and isoquinoline ring systems. The subsequent loss of a methoxy group results in one or more fragments with *m/z* 171.0682 [64]. The product ion at *m/z* 187.0628 was an isoquinoline ring, and the product ion at *m/z* 156.0443 was produced by demethylation of the fragment ion at *m/z* 171. The product ions and the corresponding neutral fragment loss noticed in our study agree with the work demonstrated by Wickens et al. (2006) [64] and Peng et al., 2007 [71] for the characteristic structural information of papaverine.

Tetrahydroisoquinoline alkaloid: The reticuline was identified at RT 15.8 min by *m/z* 192.1022, resulting from fragmentation of [M + H]^+^ 330.1701 corresponding to reticuline (C_19_H_23_NO_4_ (calculated [M + H]^+^ 330.1700). The fragmentations indicated the initial loss of methylamine to form an ion of *m/z* 299.1281, followed by fragmentation of the benzyl moiety (*m/z* 137.0600). The stable naphthalene ion (*m/z* 175.0750) was formed from the rearrangement of the ion *m/z* 299. The benzylisoquinoline ion (*m/z* 192.1022) was formed directly from the quasimolecular ion [M + H]^+^ and was the diagnostic ion for tetrahydroisoquinoline type of alkaloids [65,72].

Benzo[c]phenanthridines: Sanguinarine ([M]^+^; *m/z* 332.0917) is benzo[c]phenanthridines, most of whose fragmentation ions form by cleavage of the substituted groups rather than ring fusion. As a derivative of protopine, sanguinarine contains two methylenedioxy rings. The base peak at *m/z* 304.0969 results from the rearrangement of one methylenedioxy ring and the consequent loss of CO. Consecutive losses of CH_2_O and CO generate other fragments at *m/z* 274.0865 and 246.0915, respectively [68]. 

### 3.8. Poppy Seed Adulteration or Substitution

Poppy seeds are relatively expensive and are sometimes mixed with seeds that closely resemble poppy seeds, such as seeds of *Amaranthus* species [48,73]. The adulterants are often similar in look, color, and texture to authentic poppy seeds. Even though some adulterant seeds may not present a health hazard, they are less expensive, and their use results in economic losses for food poppy seed producers and places fraudulent products in the marketplace.

Adulterated seeds were observed under 10× to 50× magnifications, and the salient features were noted to differentiate the adulterants from the genuine poppy seeds (Figure 9 and Table 1). Poppy seeds from different cultivars vary in size, color, and appearance, with the black/blue/brown seeds having a rough waxy coat while the white seeds have a smooth non-waxy coat. Many adulterants, such as *Amaranthus* seeds, are used commercially to increase the weight of poppy seeds, leading to lower-grade materials in commerce.

In this study, an LC–ESI–MS/MS technique provided applicable information to characterize 47 compounds from *P. somniferum* and major components from adulterants using authenticated plant materials. With this methodology, ground plant material could be analyzed to confirm or deny the presence of *P. somniferum,* which should aid in detecting adulteration or preventing the use of potentially mislabeled or misidentified “*P. somniferum”* material. Poppy seeds containing varying amounts of adulterants were analyzed. Adulteration as low as 10% could be easily detected based on the constituents shown in Table 5. Other possible adulterants of poppy seeds (Table 5) are:

*Salvia hispanica*, commonly known as chia, is a species of flowering plant of the mint family, Lamiaceae, native to Central and Southern Mexico and Guatemala [74]. Ground or whole chia seeds are consumed in Paraguay, Bolivia, Argentina, Mexico, and Guatemala for nutritious drinks and as a food source. *S. hispanica* is a source of omega-3 fatty acids—linolenic acid, linoleic acid, oleic acid, pantothenic acid, myricetin, the main flavanol, quercetin, and kaempferol are found in chia seeds.

*Nigella sativa* of the Ranunculaceae family is an indigenous herbaceous plant native to Asia and the Middle East. The plant is also known as black cumin (in English) and black caraway (in the USA). It grows throughout Mediterranean countries and is cultivated extensively in India and Pakistan. In Egypt and the Middle East, the black seed oil is commonly used to relieve cough and bronchial asthma [75,76,77]. The seeds contain fixed and volatile oils, proteins, flavonoid glycosides (kaempferol 3-*O*-β-D-glucopyranosyl-(1→2)-*O*-β-D-galactopyranosyl-(1→2)-*O*-β-D-glucopyranoside), alkaloids (mainly magnoflorine), and saponins (sieboldianoside A, tauroside H2, tauroside G3, decaisoside D, sapindoside B, tauroside E). The most important active compounds were thymoquinone, thymohydroquinone, and dithymoquinone [77,78,79]. *Salvia* and *Nigella* seeds contain terpenoids and quinones that are reported to interact with opiate receptors [80].

*Amaranthus caudatus, A. paniculatus,* and *A. cruentus: Amaranthus* of the family Amaranthaceae, which includes quinoa and amaranth species, is a valuable food source of nutrients with high-quality proteins, vitamins, minerals, and bioactive compounds such as phenolics [81,82]. Approximately 35 to 55 *Amaranthus* species are native to the Americas, and at least 15 species are native to Africa, Europe, and Asia. *Amaranthus* is an important nutritional crop. It is a rare plant, with leaves consumed as a vegetable while seeds are eaten as a cereal [83,84]. *A. caudatus, A. hypochondriacus*, and *A. cruentus,* used for grain purposes, have tremendous potential to increase food production. Amaranth is commonly used in the bakery for cookies, biscuits, candies, pancakes, pasta, noodles, etc. [85]. In addition to its nutritional properties, amaranth offers an attractive source of lysine and other bioactive compounds such as phenolics, squalene, folate, phytates, and tocopherols [86]. Studies have also revealed the presence of olean triterpene saponins in the seeds.

Sesame (*Sesamum indicum*) has long been used extensively as a traditional food in the Middle East and South Asia. Sesame seed and sesame oil are widely used in cooking and as ingredients in sweet and confectionery foods. The potent antioxidant properties of seed extracts from *S. indicum* and sesame oil are attributed mainly to the presence of the lignans, including sesamin, sesamolin, sesamolinol, sesaminol, and lignan glycosides [87].

*Chenopodium quinoa*: Due to their high protein content, quinoa seeds have high nutritional value and are known to possess several flavonoids, including kaempferol glycosides, quercetin glycosides, and saponins (Table 5) [88].

*Eschscholzia californica*, the California golden poppy, California sunlight, or cup of gold, is a flowering plant in the family Papaveraceae, native to the United States and Mexico. The Californian poppy has a sap that contains a different class of alkaloids from that of the addictive “opium poppy” and does not have a narcotic effect on the human body. Similar to all poppy alkaloids, these have a sedative and relaxant effect on the body and mind, but they are perceived to be gentle and mild. Analysis showed the presence of morphinan alkaloids in opium poppy (*P. somniferum*) and benzophenanthridine alkaloids in *E. californica*. Several chemical compounds have been identified in *E. californica*, including californidine, cryptopine, eschscholtzine (californidine), sanguinarine, chelirubine, and other similar (Papaveraceae) alkaloids [89].

## 4. Conclusions

Two independent methodologies (microscopy and LC-QToF) were investigated with twenty-three poppy seed samples and eight adulterants. Detailed micromorphology observations with salient features highlighted may be used to differentiate poppy seeds from adulterant seeds. The developed LC–QToF method was identified as a sensitive and selective analytical method to determine five opiates in various colored and noncolored poppy seeds. The method was validated in terms of linearity, accuracy, selectivity, LOD, LOQ, and precision. A variation in total alkaloid content was noted among different color poppy seeds, specifically morphine, codeine, and thebaine, ranging from 0.8–223, 0.2–386, and 0.1–176 mg/kg, respectively. In most cases, the phenanthrene alkaloids morphine (dominant), codeine, and thebaine are in higher concentrations than the benzylisoquinoline alkaloids papaverine (<LOD-28 mg/kg) and noscapine (<LOD-73 mg/kg). The optimal treatment for reducing morphine and other OAs in poppy seed consists of washing, heating, and grinding. The application of LC–QToF provided useful information to characterize forty-seven compounds in poppy seed samples. Cleavage of the piperidine ring with consecutive losses of an amine and CO were some of the characteristic fragment ions for morphine, thebaine, and codeine, although this fragmentation process is not applicable for noscapine and papaverine. The developed simple and accurate method will be useful for determining and confirming unknown molecules, contaminants, or adulterants via a targeted or non-targeted approach. It is essential for identifying *P. somniferum*. There were no adulterants found in any of the poppy seed samples studied, which included seeds of amaranthus, golden poppy, black cumin, chia, sesame, and quinoa. However, the developed methods should serve as proactive tools to combat potential adulteration with seeds that closely mimic poppy seeds.

## Figures and Tables

**Figure 1 foods-12-01510-f001:**
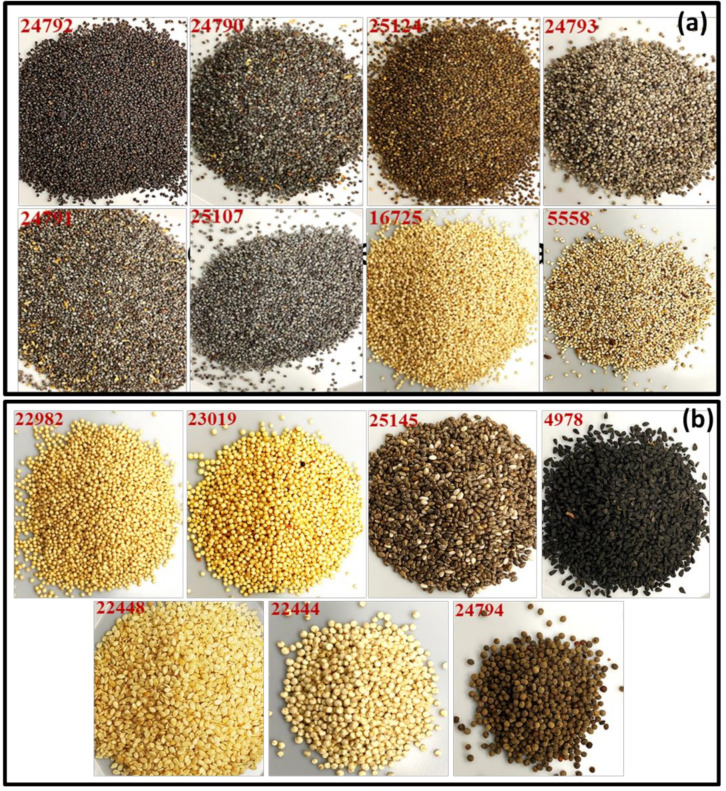
Representative images with NCNPR # of (**a**) poppy seed samples and (**b**) adulterants used in this study.

**Figure 2 foods-12-01510-f002:**
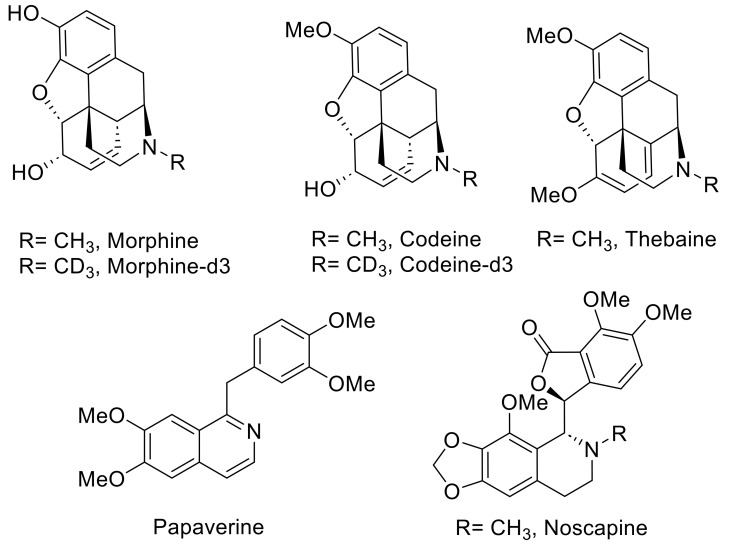
Chemical structures of five alkaloids and two internal standards, morphine-d_3,_ and codeine-d_3_.

**Figure 3 foods-12-01510-f003:**
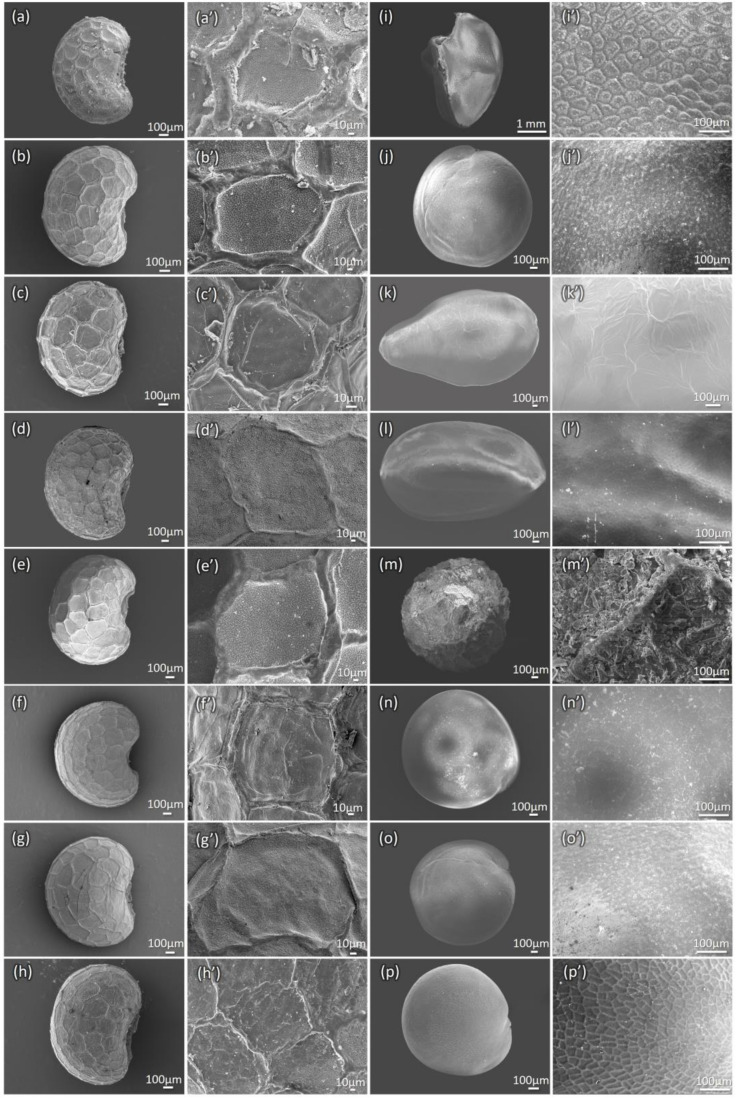
Scanning electron microscopy (SEM) observation of different poppy seeds. #24792 (**a**,**a’**); #24790 (**b**,**b’**); #25124 (**c**,**c’**); #24793 (**d**,**d’**); #24791 (**e**,**e’**); #25107 (**f**,**f’**); #16725 (**g**,**g’**); and #5558 (**h**,**h’**) show the whole seed view and testa texture with a reticulate waxy plate in blue/black/brown seeds and mild in white seeds. The adulterant samples: *N. sativa* (**i**,**i’**) has a reticulate pattern of ridges with an ocellate spinulose testa texture; *C. quinoa* (**j**,**j’**) has a “dusty” surface, with no protuberances; *S. indicum* (**k**,**k’**) has a smooth texture with some nerve lines; *S. hispanica* (**l**,**l’**) has a smooth texture; *E. californica* (**m**,**m’**) has a rough and micro sculpted texture; *A. cruentus* (**n**,**n’**), *A. caudatus* (**o**,**o’**), are with smooth texture; *A. paniculatus* (**p**,**p’**), has a reticulate somewhat hierarchically structure. Scale bars are marked respectively for each image.

**Figure 4 foods-12-01510-f004:**
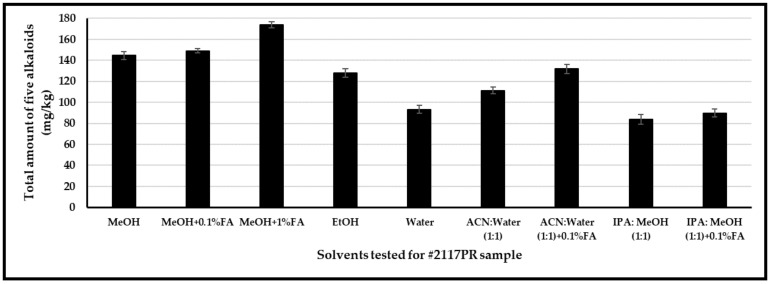
Extraction efficiency of opiates in 2117PR with various solvents.

**Figure 5 foods-12-01510-f005:**
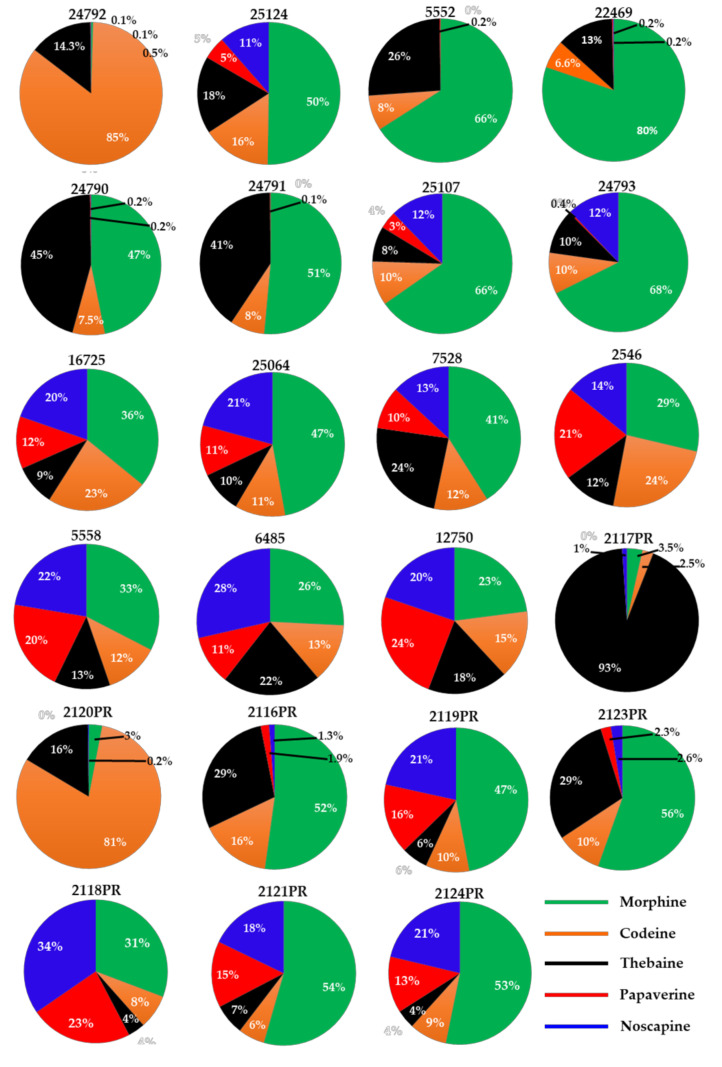
Content of alkaloids (μg/g) observed in different poppy seed samples. The area of a slice in each pie chart is proportional to the individual alkaloid’s content (μg/g).

**Figure 6 foods-12-01510-f006:**
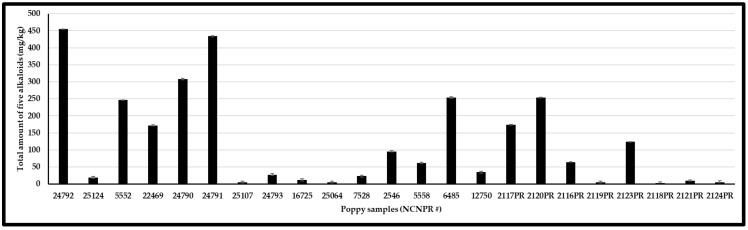
Total five OAs content (mg/kg) of different poppy seed samples.

**Figure 7 foods-12-01510-f007:**
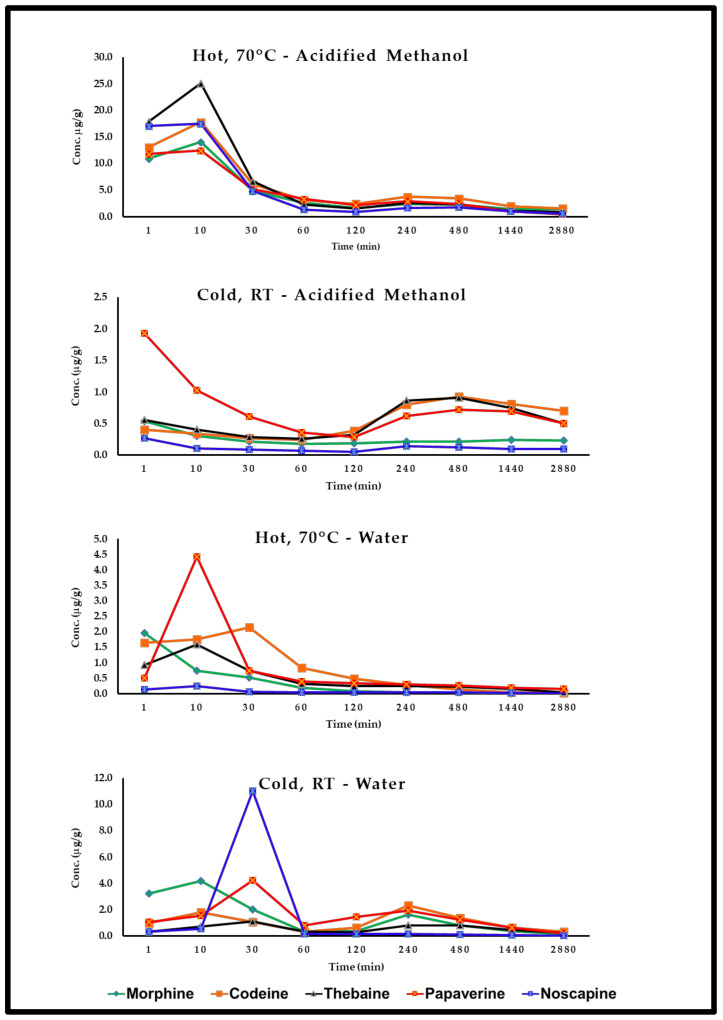
Change in alkaloid content (μg/g) during washing at different conditions and time intervals.

**Figure 8 foods-12-01510-f008:**
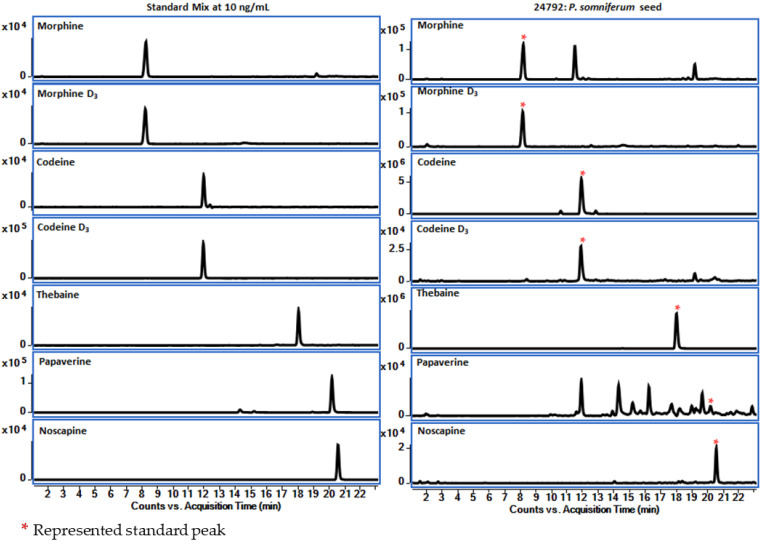
Extracted ion chromatograms of five OAs and internal standards (*) from the standard mixture and poppy seed sample (#24792).

**Figure 9 foods-12-01510-f009:**
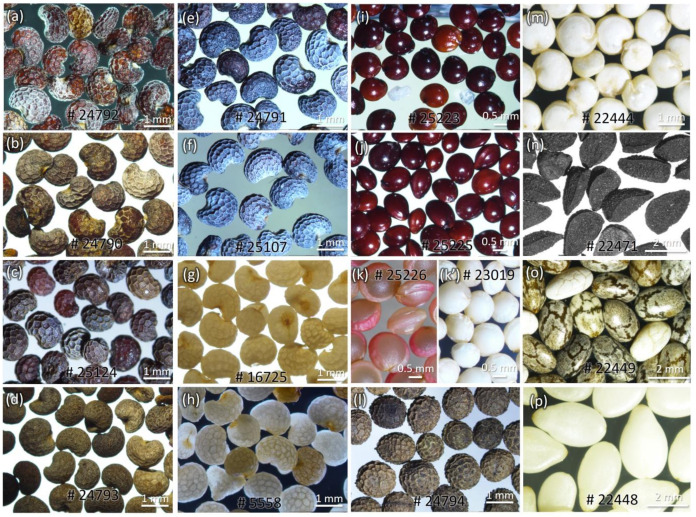
External morphology of poppy seeds and their adulterants. (**a**–**h**) *P. somniferum* (black, brown, white, and blue); (**i**) *A. paniculatus;* (**j**) *A. cruentus*; (**k**,**k’**) *A. caudatus* (pink and white, respectively); (**l**) *E. californica*; (**m**) *C. quinoa*; (**n**). *N. sativa*; (**o**) *S. hispanica*; and (**p**) *S. indicum*. Scale bar: 0.5 mm: (**i**–**k’**); 1 mm: (**a**–**h**,**l**,**m**); 2 mm: (**n**–**p**).

**Table 1 foods-12-01510-t001:** List of salient features to distinguish the poppy seeds from their adulterants.

Botanical Name	Features				
	Testa Surface	Color	Size (l × b in mm) and Shape (n = 25)	Mucilage	Hilum Location
*Papaver somniferum*	Rough, honeycomb pitted.	Blue–black/pale white	1.65 *×* 1.15 reniform	Absent	At the center of the concave curve of the seed
*Amaranthus cruentus*	Smooth, circular compressed, with a thick yellowish margin and a translucent center	White	1.54 *×* 1.461 spheroidal	Absent	Top of the seed in a notch
*Amaranthus caudatus*	Smooth, circular compressed, with a thick margin and a translucent center	White/pink	1.66 *×* 1.44 spheroidal	Absent	Top of the seed in a notch
*Amaranthus paniculatus* var*. cruentus*	Reticulate somewhat hierarchically structure	Reddish-brown	1.04 *×* 1.06 spheroidal	Absent	Top of the seed in a notch
*Chenopodium quinoa*	Very tiny dusty surface with no protuberance	Pale White	1.78 *×* 1.90 spheroidal	Absent	Hilum location is noted at radicle as hypocotyl radicle
*Eschscholzia californica*	Rough, microsculpted	Gray to gray–brown	0.5 *×* 1.5 cylindrical	Absent	On top, with a line joining both poles
*Salvia hispanica*	Smooth	Gray with black and white strips	1.95 *×* 1.23 oval	Conspicuous	At the tip of the upper end
*Sesamum indicum*	Smooth, plain white	White	4.12 *×* 2.39 flattened pear	Absent	At the tip of the upper end
*Nigella sativa*	Reticulate pattern of the ridge with ocellate spinulose.	Black	3.95 *×* 2.31 angular oblong funnel	Absent	At the tip of the upper end

l = length, b = breadth, and n = number of seeds observed and averaged.

**Table 2 foods-12-01510-t002:** LC-QToF-MS validation parameters for poppy seeds standards.

Parameters	Morphine	Codeine	Thebaine	Papaverine	Noscapine
Retention time (min)	8.2	11.9	18.0	20.2	20.6
LOD (pg/mL)	10	10	25	10	10
LOQ (pg/mL)	25	25	100	25	25
Range (ng/mL)	0.5–250	0.5–250	0.5–250	0.5–250	0.5–250
*r* ^2^	>0.999	>0.999	>0.999	>0.999	>0.999
Regression equation	*y* = 46470*x* + 21760	*y* = 60115*x* + 45412	*y* = 43224*x* − 9873	*y* = 110904*x* − 67919	*y* = 80587*x* + 5161
Precision (%RSD)					
Intra-day (%)	2.6–3.0	3.5–3.9	0.7–3.5	1.7–3.7	0.8–3.0
Inter-day (%)	3.2–3.9	3.1–3.9	1.3–3.6	2.1–3.4	1.1–2.2
Accuracy (±RSD)%	90–109% (1.7–4.4%)				

**Table 3 foods-12-01510-t003:** Content of μg/g of different alkaloids in various ground poppy seeds using LC–QToF–MS.

No.	NCNPR #	Content of Alkaloids (μg/g ± %RSD)					Total Amt. of Five Alkaloids (μg/g ± %RSD)
		Morphine	Codeine	Thebaine	Papaverine	Noscapine	
1	24792	2 ± 1 *	386 ± 3	65 ± 1	0.4 ± 1	0.4 ± 0.1	455 ± 1
2	25124	10 ± 1	3 ± 3	3 ± 4	1 ± 4	2 ± 5	19 ± 3
3	5552	161 ± 0.1	19 ± 1	63 ± 1	0.5 ± 1	DUL	246 ± 1
4	22469	138 ± 1	11 ± 4	22 ± 2	0.4 ± 0.4	0.3 ± 4	172 ± 2
5	24790	144 ± 2	23 ± 4	140 ± 2	0.5 ± 1	0.3 ± 2	309 ± 2
6	24791	223 ± 4	34 ± 3	176 ± 2	0.5 ± 0.1	DUL	434 ± 2
7	25107	3 ± 4	0.5 ± 4	0.4 ± 4	0.2 ± 2	0.6 ± 3	5 ± 4
8	24793	18 ± 2	3 ± 4	3 ± 3	0.1 ± 3	3 ± 2	27 ± 3
9	16725	4 ± 3	3 ± 4	1 ± 4	1 ± 5	2 ± 4	12 ± 4
10	25064	3 ± 4	0.6 ± 4	0.5 ± 3	0.6 ± 3	1 ± 4	5 ± 3
11	7528	9 ± 2	3 ± 4	6 ± 4	2 ± 4	3 ± 2	23 ± 3
12	2546	27 ± 1	23 ± 1	11 ± 2	20 ± 3	14 ± 1	96 ± 2
13	5558	20 ± 1	8 ± 3	8 ± 3	13 ± 3	14 ± 1	62 ± 2
14	6485	65 ± 1	33 ± 0.5	55 ± 3	28 ± 4	73 ± 3	254 ± 2
15	12750	8 ± 3	5 ± 4	6 ± 4	8 ± 4	7 ± 2	34 ± 3
16	2117PR	6 ± 2	4 ± 4	162 ± 1	DUL	2 ± 0.2	174 ± 2
17	2120PR	8 ± 3	205 ± 1	41 ± 3	DUL	0.5 ± 0.4	254 ± 2
18	2116PR	33 ± 1	10 ± 0.3	19 ± 1	1 ± 0.4	0.8 ± 1	64 ± 2
19	2119PR	2 ± 4	0.5 ± 4	0.3 ± 4	0.8 ± 3	1 ± 3	5 ± 4
20	2123PR	69 ± 0.4	13 ± 1	36 ± 0.2	3 ± 0.4	3 ± 1	124 ± 1
21	2118PR	0.8 ± 4	0.2 ± 4	0.1 ± 4	0.6 ± 4	0.9 ± 3	3 ± 4
22	2121PR	6 ± 2	0.6 ± 2	0.7 ± 2	2 ± 3	2 ± 2	10 ± 2
23	2124PR	3 ± 3	0.4 ± 4	0.2 ± 4	0.6 ± 3	1 ± 4	5 ± 4

* Mean values (n = 3) ± %RSD; DUL = detected under limits of quantification.

**Table 4 foods-12-01510-t004:** RT, compound name, molecular formula, mass, *m/z,* and major fragment ions for reference compounds and other individual compounds detected in poppy seed samples using LC–QToF.

No.	RT (min)	Compounds	Formula	Mass	[M + H]^+^ (*m/z*)	Fragment Ions of MS-MS	Class of Compounds
1	3.3	Dopamine	C_8_H_11_NO_2_	153.0790	154.0863 (154.0863) *	137.0598 [M + H-NH_3_]^+^, 119.0488 [M + H-NH_3-_H_2_O]^+^, 109.0647, 91.0543, 65.0389	Catecholamine
2	2.91	Phenylalanine	C_9_H_11_NO_2_	165.0790	166.0863 (166.0863)	120.0809 [M + H-H_2_O-CO]^+^, 103.0541 [M + H-H_2_O-CO-NH_3_]^+^	Amino acid
3	4.5	Leucine/Isoleucine	C_6_H_13_NO_2_	131.0946	132.1019 (132.1019)	86.0966 [M + H-H_2_O-CO]^+^, 69.0696 [M + H-H_2_O-CO-NH_3_]^+^	
4	4.94						
5	6.6	Adenosine	C_10_H_13_N_5_O_4_	267.0968	268.1042 (268.1040)	136.0619 [M + H-132]^+^, 119.0351 [M + H-132-NH_3_]^+^, 94.0400 [M + H-132-NH_2_CN]^+^	Purine nucleoside
6	7.05	Guanosine	C_10_H_13_N_5_O_5_	283.0917	284.0890 (284.0989)	152.0567 [M + H-132]^+^, 135.0300 [M + H-152-NH_3_]^+^, 107.0489, 110.0345 [M + H-152-NH_2_CN]^+^	
7	8.2	Morphine	C_17_H_19_NO_3_	285.1365	286.1440 (286.1438)	229.0854 [M + H-CH_2_CHNHCH_3_]^+^, 211.0783 [M + H-CH_2_CHNHCH_3_-H_2_O]^+^, 201.0910 [M + H-CH_2_CHNHCH_3_-CO]^+^, 173.0596 [M + H-CH_2_CHNHCH_3_-2CO]^+^, 183.0804 [M + H-CH_2_CHNHCH_3_-CO-H_2_O]^+^, 185.0596 [M + H-CH_2_CHNHCH_3_-C_2_H_4_O]^+^, 165.0701 [M + H-CH_2_CHNHCH_3_-CO-2H_2_O]^+^, 155.0856 [M + H-CH_2_CHNHCH_3_-2CO-H_2_O]^+^, 153.0699, 58.0657 [C_3_H_8_N]^+^	Morphinane alkaloid
8	8.45	Morphinone	C_17_H_17_NO_3_	283.1208	284.1284 (284.1281)	227.0708 [M + H-CH_2_CHNHCH_3_]^+^, 209.0597 [M + H-CH_2_CHNHCH_3_-H_2_O]^+^, 199.0752 [M + H-CH_2_CHNHCH_3_-CO]^+^, 185.0598 [M + H-CH_2_CHNHCH_3_-CH_2_CO]^+^, 58.0656	
9	13.2	Codeinone	C_18_H_19_NO_3_	297.1365	298.1441 (298.1438)	241.1091 [M + H-CH_2_CHNHCH_3_], 239.0705 [M + H-CH_2_CHNHCH_3_], 223.0765 [M + H-C_2_H_5_NHCH_3_], 213.0547 [M + H-CH_2_CHNHCH_3_-CO], 223.0765 [M + H-C_2_H_5_NHCH_3_-CO], 199.0717 [M + H-CH_2_CHNHCH_3_-CH_2_CO], 58.0655	
10	18.0	Thebaine	C_19_H_21_NO_3_	311.1521	312.1597 (312.1594)	281.1176 [M + H-CH_3_NH_2_], 266.0944 [M + H-CH_3_NH_2_-CH_3_], 251.0706 [M + H-CH_3_NH_2_-2CH_3_], 249.0917 [M + H-CH_3_NH_2_-CH3OH], 58.0656 [C_3_H_8_N]^+^	
11	11.9	Codeine	C_18_H_21_NO_3_	299.1521	300.1596 (300.1594)	243.1019 [M + H-CH_2_CHNHCH_3_]^+^, 225.0909 [M + H-CH_2_CHNHCH_3_-H_2_O]^+^, 215.1067 [M + H-CH_2_CHNHCH_3_-CO]^+^, 183.0807 [M + H-CH_2_CHNHCH_3_-CH_3_OH]^+^, 199.0751[M + H-CH_2_CHNHCH_3_-C_2_H_4_O]^+^, 165.0698 [M + H-CH_2_CHNHCH_3_-CH_3_OH-H_2_O]^+^, 155.0855 [M + H-CH_2_CHNHCH_3_-CH3OH-CO]^+^, 58.0655	
12	10.8	Neopine	C_18_H_21_NO_3_	299.1521	300.1597 (300.1594)	243.1027 [M + H-CH_2_CHNHCH_3_]^+^, 225.0911 [M + H-CH_2_CHNHCH_3_-H_2_O]^+^, 199.0754, 171.0806, 165.0699, 141.0698, 58.0659	
13	7.2	Morphine-*N*-Oxide isomers	C_17_H_19_NO_4_	301.1314	302.1390 (302.1387)	286.1443 [M + H-16]^+^, 229.0854 [M + H-16-CH_2_CHNHCH_3_]^+^, 266.1180, 215.0941, 201.0910 [M + H-16-CH_2_CHNHCH_3_-CO]^+^, 165.0695 [M + H-16-CH_2_CHNHCH_3_-CO-2H_2_O]^+^, 77.0389, 58.0655	
14	8.5						
15	10.4						
16	13.5	Morphine; (-)-form, Di-Me ether	C_19_H_23_NO_3_	313.1678	314.1756 (314.1751)	299.1551 [M + H-CH_3_]^+^, 58.0656	
17	11.4	Codeine; *N*-Oxide	C_18_H_21_NO_4_	315.1471	316.1544 (316.1543)	300.1593, 58.0654	
18	11.6						
19	13.6						
20	13.1	Thebaine; (-)-form, O3-De-Me	C_18_H_19_NO_3_	297.1365	298.1440 (298.1438)	267.1026 [M + H-CH_3_NH_2_]^+^, 252.0777 [M + H-CH_3_NH_2_-CH_3_]^+^, 237.0555 [M + H-CH_3_NH_2_-2CH_3_]^+^, 58.0656	
21	14.69	Thebaine-*N*-oxide isomers	C_19_H_21_NO_4_	327.1471	328.1551 (328.1543)	-	
22	16.44						
23	16.6	Narcotine; (1R,9S)-form, O8-De-Me	C_21_H_21_NO_7_	399.1318	400.1387 (400.1391)	372.1442, 206.0816, 178.0864, 163.0630, 58.0657	
24	17.8	Tetrahydropapaverine	C_20_H_25_NO_4_	343.1784	344.1860 (344.1856)	340.1545, 313.1450 [M + H-CH_3_NH_2_]^+^, 151.0754 [C_9_H_11_O_2_]^+^, 137.0599 [C_8_H_9_O_2_]^+^	Benzylisoquinoline alkaloid
25	17.7	Papaveroline; 4′,6,7-Tri-Me ether	C_19_H_19_NO_4_	325.1314	326.1390 (326.1387)	310.1077 [M + H-CH_4_]^+^, 188.0708 [M + H-122-16]^+^, 156.0443, 128.0495, 58.0656	
26	18.7						
27	19.3	Laudanosine; (S)-form (*N*-Methyl-1,2,3,4-tetrahydropapaverine)	C_21_H_27_NO_4_	357.1940	358.2018 (358.2013)	342.1345 [M + H-CH_4_]^+^, 206.1177, 189.0911 [M + H-122-16-OCH_3_]^+^,151.0754, 58.0655	
28	19.9	Papaveraldine	C_20_H_19_NO_5_	353.1263	354.1340 (354.1336)	151.0757, 188.0706, 58.0655	
29	20.2	Papaverine	C_20_H_21_NO_4_	339.1471	340.1541 (340.1543)	324.1236 [M + H-CH_4_]^+^, 308.0924, 280.0976, 202.0868 [M + H-122-16]^+^, 187.0628, 171.0682 [M + H-122-16-OCH_3_]^+^, 156.0443	
30	20.6	Narcotine; (1R,9S)-form (Noscapine)	C_22_H_23_NO_7_	413.1475	414.1545 (414.1547)	353.1001 [M + H-aziridine(43Da)-H_2_O]^+^, 220.0972 [M + H-C_10_H_10_O_4_]^+^, 205.0736 [M + H-C_10_H_10_O_4_-CH_3_]^+^, 188.0705 [M + H-C_10_H_10_O_4_-CH_3_-OH]^+^, 179.0716 [M + H-C_10_H_10_O_4_-41]^+^, 175.0655 [M + H-C_10_H_10_O_4_-CH_3_-OCH_2_]^+^	Phthalideisoquinoline
31	15.6	Glaucamine; 8-Epimer, O2-de-Me, O8-Me (*N*-Methylporphyroxine/*N*-Methylpapaver-rubine D)	C_21_H_23_NO_6_	385.1525	386.1595 (386.1598)	209.0786, 156.0523, 140.0680, 77.0387, 67.0544, 58.0655	Rhoeadine alkaloid
32	16.5	Glaucamine; O2-De-Me (*N*-Methyl-14-*O*-demethylepipor-phyroxine)	C_20_H_21_NO_6_	371.1369	372.1440 (372.1442)	322.1074, 307.0841, 279.0889, 262.0631, 178.0864, 163.0628, 91.0543, 58.0655	
33	18.1	Rhoeadine	C_21_H_21_NO_6_	383.1369	384.1445 (384.1442)	368.1129 [M-15]^+^, 177.0786	
34	15.8	Reticuline; (S)-form	C_19_H_23_NO_4_	329.1627	330.1701 (330.1700)	299.1281 [M + H-CH_3_NH_2_]^+^, 192.1022 [M + H-C_8_H_9_O_2_]^+^, 175.0750, 177.0787[M + H-C_8_H_9_O_2_-CH_3_]^+^, 143.0491, 137.0600 [M + H-CH_3_NH_2_-C_10_H_10_O_2_]^+^, 115.0542, 91.0544	Tetrahydroisoquinoline alkaloid
35	17.3	Laudanidine; (R)-form	C_20_H_25_NO_4_	343.1784	344.1851 (344.1856)	137.0598	
36	9.2	N-methyl-2,3-dioxole-tetrahydroisoquinoline	C_11_H_13_NO_2_	191.0946	192.1017 (192.1019)	177.0783, 148.0756	
37	12.8	6,7-dimethoxy-2-methyl-1,2,3,4-tetrahydroisoquinoline	C_12_H_17_NO_2_	207.1259	208.1330 (208.1332)	58.0655	
38	14.2	Hydrocotarnine	C_12_H_15_NO_3_	221.1052	222.1121 (222.1125)	118.0860, 114.0914, 77.0388, 58.0655	
39	14.95	Salutaridine	C_19_H_21_NO_4_	327.1471	328.1540 (328.1543)	297.1132 [M + H-CH_3_NH_2_]^+^, 298.1430 [M + H-CH_2_O]^+^, 265.0857 [M + H-CH_3_NH_2_-CH_3_OH]^+^, 253.0860 [M + H-CH_2_CHNHCH_3_-H_2_O]^+^, 239.0705 [M + H-CH_2_CHNHCH_3_-CH_3_OH]^+^, 237.0905 [M + H-CH_3_NH_2_-CH_3_OH-CO]^+^, 233.0597 [M + H-CH_3_NH_2_-2CH_3_OH]^+^, 221.0601 [M + H-CH_2_CHNHCH_3_-CH_3_OH-H_2_O]^+^, 211.0756 [M + H-CH_2_CHNHCH_3_-CH_3_OH-CO]^+^, 207.0805 [M + H-CH_2_CHNHCH_3_-2CH_3_OH]^+^, 205.0649 [M + H-CH_3_NH_2_-2CH_3_OH-CO]^+^, 181.0649 [M + H-CH_2_CHNHCH_3_-2CH_3_OH-CO]^+^	Pro-Morphinane alkaloid
40	15.8	Salutaridinol	C_19_H_23_NO_4_	329.1627	330.1703 (330.1700)	298.1455 [M + H-CH_3_OH]^+^, 267.1020 [M + H-CH_3_NH_2_-CH_3_OH]^+^, 255.1020 [M + H-CH_2_CHNHCH_3_-H_2_O]^+^, 241.0861 [M + H-CH_2_CHNHCH_3_-CH_3_OH]^+^, 239.1067 [M + H-CH_3_NH_2_-CH_3_OH-CO]^+^, 223.0757 [M + H-CH_2_CHNHCH_3_-CH_3_OH-H_2_O]^+^, 213.0904 [M + H-CH_2_CHNHCH_3_-CH_3_OH-CO]^+^, 209.0602 [M + H-CH_2_CHNHCH_3_-2CH_3_OH]^+^, 207.0808 [M + H-CH_3_NH_2_-2CH_3_OH-CO]^+^, 192.1022 [M + H-CH_2_CHNHCH_3_-2CH_3_OH-2H_2_O]^+^ [C_11_H_14_NO_2_]^+^, 181.0651 [M + H-CH_2_CHNHCH_3_-2CH_3_OH-CO]^+^	
41	18.73	Cryptopine; 13-Oxo	C_21_H_21_NO_6_	383.1369	384.1441 (384.1442)	310.1076, 241.1051, 91.0543, 58.0655	Protopine
42	19.4	Cryptopine	C_21_H_23_NO_5_	369.1576	370.1650 (370.1649)	291.1013, 222.1109, 204.1020, 165.0908, 149.0596, 58.0656	
43	19.8	Protopine	C_20_H_19_NO_5_	353.1263	354.1338 (354.1336)	336.1206 [M + H-H_2_O]^+^, 323.0939 [M + H-CH_3_NH_2_], 206.0808, 188.0708 [M + H-H_2_O-148(RDA)]^+^, 149.0597, 58.0655	
44	19.0	Protopine; 14-Alcohol	C_20_H_21_NO_5_	355.142	356.1492	322.1076, 294.1125, 140.0683, 58.0655	
45	20.9	Allocryptopine	C_21_H_23_NO_5_	369.1576	370.1648 (370.1649)	206.0812, 188.0707, 149.0595, 58.0657	
46	21.4	Narceine	C_23_H_27_NO_8_	445.1737	446.1807 (446.1809)	428.1707 [M + H-18]^+^, 383.1127 [M + H-63]^+^, 365.1023 [M + H-81]^+^, 350.0789, 190.0864, 58.0656	Stilbene
47	23.01	Sanguinarine	C_20_H_14_NO_4_^+^	332.0923	332.0925 (332.0923)	304.0969 [M + H-CO]^+^, 317.0685 [M + H-CH_3_]^+^, 302.0811 [M + H-CH_2_O]^+^, 274.0865 [M + H-2H-2CO]^+^, 272.0708 [M + H-2H-2CO-2H]^+^, 246.0915 [M + H-2H-3CO]^+^	Benzophenanthridine

* Theoretical mass; -CH3NH2 = 31.0422 Da; -CH3OH = 32.0262 Da; -CO = 27.9949; -CH2CHNHCH3 = 57.0579 Da; -H2O = 18.0106 Da; -CH3 = 15.0235; -CH2O = 30.0106 Da; -OCH3 = 31.0184 Da; -CH4 = 16.0313 Da; -OCH2 = 30.0106 Da; -NHCH2 = 29.0265 Da; -NHCO = 43.0058 Da; NH3 = 17.0265 Da; -CH2CH2 = 28.0313 Da; -HCHO = 30.0106 Da.

**Table 5 foods-12-01510-t005:** Adulterants derived from plants in poppy seed adulteration: plant name, compound name, retention time in minutes, molecular formula, mass, *m/z*, and major ion fragments for adulterants detected using LC–QToF.

Plant Name	Compound Name	RT (min)	Molecular Formula	Mass	[M + H]^+^/[M + Na]^+^ (*m/z*)
*Nigella sativa* (black caraway or black cumin seeds) (#4978, 7516)	Magnoflorine	15.2	C_20_H_24_NO_4_^+^	342.1705	342.1705 [M]^+^
	Thymoquinone	20.03	C_10_H_12_O_2_	164.0837	165.091
	kaempferol 3-*O*-β-D-glucopyranosyl-(1→2)-*O*-β-D-galactopyranosyl-(1→2)-*O*-β-D-glucospyranoside	15.8	C_33_H_40_O_21_	772.2062	773.2135/795.1954
	Tauroside E	34.3	C_41_H_66_O_12_	750.4554	751.4627/773.4446
	Sapindoside B	33.6	C_46_H_74_O_16_	882.4977	883.505/905.4869
*Sesamum indicum* (#2519, 22448)	Sesamin	35.5	C_20_H_28_O_6_	364.1886	365.1959/387.1778
	Sesamolin	30.55	C_20_H_28_O_7_	380.1835	381.1908/403.1727
	Sesamolinol; *O*-β-D-glucopyranoside	25.5	C_26_H_30_O_12_	534.1737	535.181/557.1629
	Sesaminol triglucoside	19.6	C_38_H_48_O_22_	856.2637	857.271/879.2529
	Sesamolinol gentiobioside	23.2	C_32_H_40_O_17_	696.2265	697.2338/714.2604/719.2158
*Salvia hispanica* (Chia seeds) (#22449)	8,13-Epoxy-15,18-labdanedioic acid; (8α,13*S*)-form	31.01	C_20_H_32_O_5_	352.2250	353.2323
	15,16-Epoxy-10-hydroxy-13(16),14-clerodadiene-17,12:18,19-diolide	24.7	C_20_H_22_O_6_	358.1416	359.1489/381.1309
	Caffeic acid	13.8	C_9_H_8_O_4_	180.0423	179.035 [M-H]^−^
	Linolenic acid	35.0	C_18_H_30_O_2_	278.2246	277.2173 [M-H]^−^
	Hydroxylinolenic acid	36.7/38.3	C_18_H_30_O_3_	294.2195	293.2122 [M-H]^−^
	Linoleic acid	40.8	C_18_H_32_O_2_	280.2402	279.233 [M-H]^−^
	Oleic acid	40.9	C_18_H_34_O_2_	282.2559	281.2486 [M-H]^−^
	Rosmarinic acid	18.2/20.3	C_18_H_16_O_8_	360.0845	359.0772 [M-H]^−^
	Salvipalestinoic acid	20.9	C_20_H_24_O_7_	376.1522	377.1595/399.1414
	Salviaflaside	18.2	C_24_H_26_O_13_	522.1373	523.1446/540.1712/545.1266 (frag. 163.0393)
	Sepulturin F	19.5/19.8	C_20_H_22_O_7_	374.1366	375.1438/397.1258 (frag. 357.1335)
	Niacin	2.8	C_6_H_5_NO_2_	123.032	124.0393
*Amaranthus caudatus* (Kiwicha seeds) (#23019, 25226)	Linoleic acid		C_18_H_32_O_2_	280.2402	
	Leucine/isoleucine	4.5/4.9	C_6_H_13_NO_2_	131.0946	132.1019
	Hydroxylinolenic acid	36.7/38.3	C_18_H_30_O_3_	294.2195	293.2122 [M-H]^−^
	2,3,23-Trihydroxy-30-nor-12,20(29)-oleanadien-28-oic acid; (2β,3β)-form, 23-Carboxylic acid, 28-O-β-D-glucopyranosyl ester	23.8	C_35_H_52_O_11_	648.3510	647.3437 [M-H]^−^
	2,3,6,23-Tetrahydroxy-12-oleanen-28-oic acid; (2β,3β,6α)-form, 23-Aldehyde, 3-O-[α-L-rhamnopyranosyl-(1→2)-β-D-glucopyranoside], 28-O-β-D-glucopyranosyl ester	25.5	C_48_H_76_O_20_	972.4930	971.4857 [M-H]^−^
*Amaranthus cruentus* (purple or red amaranthus) (#22982, 25225)	Linoleic acid		C_18_H_32_O_2_	280.2402	
	Leucine/isoleucine	4.5/4.9	C_6_H_13_NO_2_	131.0946	132.1019
	Hydroxylinolenic acid	36.7/38.3	C_18_H_30_O_3_	294.2195	293.2122 [M-H]^−^
	HydroxyLinoleic acid	38.1/38.4	C_18_H_32_O_3_	296.2351	297.2424 [M-H]^−^
	Amaranthus saponin I	28.4	C_48_H_76_O_19_	956.4981	955.4880 [M-H]^−^
	Amaranthus saponin II	26.7	C_48_H_74_O_20_	970.4773	969.4701 [M-H]^−^ (Frag. 807.4148 [M-H-Glc (162)]^−^, 193.0356)
	3-β-*O*-[α-L-rhamnopyranosyl(1 → 3)-β-glucuronopyranosyl]-2β,3β,23-trihydroxyolean-12-en-28-oic acid 28-*O*-[β-D-glucopyranosyl] ester	25.56	C_48_H_76_O_20_	972.4930	971.4857 [M-H]^−^
	28-β-D-glucopyranosyl (2β,3β,4α)-3-(β-D-glucopyranuronosyloxy)-2-hydroxy-30-noroleana-12,20(29)-diene-23,28-dioate	23.87	C_41_H_60_O_17_	924.3831	823.3758 [M-H]^−^ (Frag. 647.3418 [M-H-Glu (176)]^−^, 193.0356)
	30-noroleana-12,20(29)-diene-23,28-dioic acid, 2,3-dihydroxy-(2β,3β,4α)-	39.9	C_29_H_42_O_6_	486.2981	485.2909 [M-H]^−^
	2,3-dihydroxy-12-oleanene-23,28-dioic acid; 3-*O*-β-D-glucuronopyranoside, 28-*O-*β-D-glucopyranosyl ester	25.9	C_42_H_64_O_17_	840.4144	839.4071[M-H]^−^ (Frag. 663.3724 [M-H-Glu (176)]^−^, 193.0356)
*Amaranthus paniculatus* (purple or red amaranthus) (#25223)	Leucine/isoleucine	4.5/4.9	C_6_H_13_NO_2_	131.0946	132.1019
	hydroxybenzoic acid	11.69	C_7_H_6_O_3_	138.0317	137.0244 [M-H]^−^
	Amaranthus saponin III	26.7	C_47_H_72_O_19_	940.4668	939.4595 [M-H]^−^
	Tetrahydroxy-oleanenoic acid; form, Aldehyde, *O*-rhamnopyranosyl-D-glucopyranoside] -D-glucopyranosyl ester	25.5	C_48_H_76_O_20_	972.493	971.4857 [M-H]^−^
	Trihydroxy-30-nor-oleanadien-28-oic acid; -form, Carboxylic acid, glucuronopyranoside, *O*-D-glucopyranosyl ester	23.9	C_41_H_60_O_17_	824.3831	823.3758 [M-H]^−^
	Amaranthus saponin IV	24.78	C_47_H_70_O_20_	954.4460	953.4388 [M-H]^−^
	Amaranthus saponin II	26.75	C_48_H_74_O_20_	970.4773	969.4701 [M-H]^−^
*Eschscholzia californica* (California poppy) (#24794) benzophenanthridine alkaloids	Californidine	20.1	C_20_H_20_NO_4_^+1^	338.1392	338.1394 (frag. 293.0811, 263.0706, 235.0759, 205.0652, 177.0701)
	Escholtzine	16.7	C_19_H_17_NO_4_	323.1158	324.1231 (frag. 187.0627)
	*N-*methyllaurotetanine	18.4	C_20_H_23_NO_4_	341.1627	342.1702 (frag. 237.0847)
	Caryachine	14.1/16.4	C_19_H_19_NO_4_	325.1314	326.1387 (frag. 223.0755, 205.0650)
	Protopine/chelidonine	19.4	C_20_H_19_NO_5_	353.1263	354.1336 (frag. 237.0791)
	Cryptopine	20.3	C_21_H_23_NO_5_	369.1576	370.1649
	Sanguinarine	22.4	C_20_H_14_NO_4_^+^	332.0923	332.0929 (frag.317.0687, 274.0865, 246.0918, 218.0965)
	Eschscholtzidine/ *O*-methylcaryachine	19.1	C_20_H_21_NO_4_	339.1471	340.1543
*Chenopodium quinoa* (#6146, 22444) * (flavonoids and saponins)	Kaempferol 3-*O*-[α-L-rhamnopyranosyl (1″–2″)]-β-D-galactopyranoside	16.6	C_27_H_30_O_15_	594.1585	595.1657 (frag.449.1084, 287.0555)
	Kaempferol*-3-O-*(apiofuranosyl*-*(1‴*-*2″))*-*galactopyranoside	17.8	C_26_H_28_O_15_	580.1428	581.1501 (frag. 449.1080, 287.0552)
	Kaempferol 3-*O*-[β-D-apiofuranosyl (1′–2″)-α-L-rhamnopyranosyl (1″–6″)]-β-D-galactopyranoside	17.01	C_32_H_38_O_19_	726.2007	727.2080 (frag. 595.1655, 449.1079, 287.0559)
	Quercetin 3-rhamninoside	15.65	C_33_H_40_O_20_	756.2113	757.2186 (frag. 611.1610, 465.1027, 303.0502)
	Quercetin 3-*O*-[β-D-xylosyl-(1->2)-β-D-glucoside]	16.76	C_26_H_28_O_16_	596.1377	597.1450 (frag. 465.1029, 303.0506)
	quercetin 3-*O*-[β-D-apiofuranosyl (1′–2″)-α-L-rhamnopyranosyl (1″–6″)]-β-D-galactopyranoside	16.08	C_32_H_38_O_20_	742.1956	743.2029 (frag. 463.1430, 303.0498)
	Kaempferol 3-rhamninoside	16.5	C_33_H_40_O_19_	740.2164	741.2237 (frag. 595.1662, 449.1082, 287.0555)
	quercetin 3-*O-*(2,6-di-α-L-rhamnopyranosyl)-β-D-galactopyranoside	15.69/ 16.09	C_27_H_30_O_16_	610.1534	611.1607 (frag. 465.1032, 303.0505)
	Hederagenin bisdesmosides; Triglycosides, 3-*O*-[β-D-Xylopyranosyl-(1→3)-β-D-glucuronopyranoside], 28-*O*-β-D-glucopyranosyl ester	26.9	C_47_H_74_O_19_	942.4824	941.4752 [M-H]^−^
	3,27-Dihydroxy-12-oleanen-28-oic acid; 3β-form, 27-Aldehyde, 3-*O*-[β-D-glucopyranosyl-(1→3)-α-L-arabinopyranoside], 28-*O*-β-D-glucopyranosyl ester	29.4	C_47_H_74_O_18_	926.4875	925.4802 [M-H]^−^
	3-Hydroxy-12-oleanene-28,30-dioic acid; 3β-form, 30-Me ester, 3-*O*-α-L-arabinopyranoside, 28-*O*-β-D-glucopyranosyl ester	29.9	C_42_H_66_O_14_	794.4453	793.4380 [M-H]^−^
	3-Hydroxy-12-oleanene-28,30-dioic acid; 3β-form, 30-Me ester, 3-*O*-[β-D-glucopyranosyl-(1→3)-α-L-arabinopyranoside], 28-*O*-β-D-glucopyranosyl ester	28.1	C_48_H_76_O_19_	956.4981	955.4908 [M-H]^−^
	3-Hydroxy-12-oleanene-28,30-dioic acid; 3β-form, 30-Me ester, 3-*O*-[β-D-glucopyranosyl-(1→2)-β-D-glucopyranosyl-(1→3)-α-L-arabinopyranoside], 28-*O*-β-D-glucopyranosyl ester	26.6	C_54_H_86_O_24_	1118.5509	1117.5436 [M-H]^−^

## Data Availability

The data presented in this study are available on request from the corresponding author.

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
