# Peer review of "Applicability of LC-QToF and Microscopical Tools in Combating the Sophisticated, Economically Motivated Adulteration of Poppy Seeds"

_foods, 2023, doi:10.3390/foods12071510_

Round 1
Reviewer 1 Report
In this study the authors investigated the use of macroscopy and LC-QToF methodologies for the identification and characterization of poppy seed samples and adulterants. Macromorphological observations and the salient characteristics of the poppy seeds were identified and differentiated from adulterants. The LC-QToF method was found to be sensitive and selective in determining five opiates in colored and noncolored poppy seeds. The method was validated and used to identify and characterize 47 compounds in the poppy seed samples. The study found a variation in the total alkaloid content between different color poppy seeds, with phenanthrene alkaloids in higher concentrations than benzylisoquinoline alkaloids. The optimal treatment for reducing morphine and other alkaloids in poppy seeds consists of washing, heating, and grinding. The developed method is useful for determining and confirming unknown molecules, contaminants, or adulterants/substituents and is relevant in identifying P. somniferum. No adulterants were found in the poppy seed samples.
In my opinion, efforts in the search for knowledge that contributes to the development of a new LC-QToF method was found to be sensitive and selective in this context, and the present work is interesting and valuable. The manuscript is well written and the experiments were carefully designed and carried out. The overall experimental approach, analyses, and presentation of the data are adequate and satisfactory supporting the conclusions made in this study. This manuscript, in my opinion, in its current form following the incorporation of a few minor suggestions.
In References section sometimes the abbreviated name of the journal is used, please correct providing consistently the full name of the journal.
In References section the Authors sometimes used the style “sentence case” or “as Title Case” – please make this consistent.
Author Response
Reviewer 1 Comments:
- In References section sometimes the abbreviated name of the journal is used, please correct providing consistently the full name of the journal.
- In References section the Authors sometimes used the style “sentence case” or “as Title Case” – please make this consistent.
Response (1-2): The references were formatted per journal guidelines, and consistency was maintained throughout references with sentence cases as suggested by the reviewer.
Reviewer 2 Report
The manuscript is very interesting because currently there are more and more adulterations of food products to reduce costs, which can affect people's health and be harmful. However, I do not see much consistency in the experiments carried out and the results and conclusions obtained. For these reason, I list below some aspects and questions that have not been clear to me for further explanation, as well as some comments and opinions.
Introduction: The introduction is too long, it should be shortened a bit by mentioning only the most important aspects. In addition, there is some information, such as the factors responsible for variable amounts of opiates, that is mentioned twice, that is, it is in two different paragraphs.
Lines 119-130: Add some references to justify these statements.
Lines 151-156: Everything that has been done in this work is mentioned, but I am missing a paragraph indicating the importance of carrying it out, focusing a little more on why it is going to be done. Emphasize the objective of the work and the main contribution it makes.
Materials and Methods: Line 206: I do not quite understand what you mean. For the second and third repetition, the appropriate amount of sample was weighed again, the solvent was added and it was sonicated or the initially weighed amount was re-extracted a second time with a new volume of solvent?
Section 2.8.2: I do not quite see the difference between the conclusions that would be obtained with the experiments carried out in each of the two paragraphs.
Results and Discussion: Section 3.2.: It would be interesting to include a figure such as a bar chart with the amount obtained with each of the solvents used and their corresponding error bars to see if there are significant differences between them and to be able to find the differences between all the possibilities of solvents analyzed.
Section 3.3.: When developing the chromatographic method, was only the mobile phase optimized? Were different gradients, column temperature, solvent flows also tried to get the most optimal values?
Figure 5: The error bars of each seed lot obtained from the two replicates should be included. Moreover, it would be convenient to indicate by means of equal or different letters the existence or not of significant differences between all the different bars of the graph.
Lines 398-409: This shows the results obtained descriptively, explaining whats is observed in the table. A greater deeper discussion explaining if the results obtained make sense and are consistent, with a greater comparison with the existing literature would be convenient. Have similar concentrations to those reported in the literature been obtained for similar samples?
Sections 3.6.1 and 3.6.2.: An inadequate cleaning of the seeds seems to be responsible for the variability in the concentration. However, when washing the seed, the alkaloids to be determined are eliminated. Therefore, how could uniformity in the measurements be obtained so that it was reproducible and conclusions could be make?
Lines 602-608: This would fit better at the end of the introduction as the objective and importance of the work to be performed.
Conclusions: It is mentioned that only 8 adulterants were analyzed. However, Table 5 lists many more adulterants identified. What is it referring to in each case?
In order to study the compounds coming from the adulteration, this adulteration was carried out by the researchers of the work to see the effect that it produces. However, did none of the seeds analyzed show adulteration? Does this adulteration occur after the harvest, before it is marketed, or when?
Additionally, you must take care of some formal aspects of the manuscript:
Lines 165, 183, 201, 222,…: Capitalize each word according to the format of the journal. Unify and apply to the entire manuscript.
Lines 320 – 328: Review the format of the paragraph, it should not be in bold. Put it in normal and justified font.
Put a separation after and before “–”, “=”, and “<”. Unify and apply to the entire manuscript.
Figure 5: The name of the axes is missing.
Check the references according to the format of the journal. The name of the journal would be always in abbreviated format and italics, and after each abbreviated word there should be a dot “.”.
References 1, 3, 25, 43, 57, 82: The DOI is missing.
References 62, 78: The page number is missing.
Author Response
Reviewer 2 Comments:
Introduction:
- The introduction is too long, it should be shortened a bit by mentioning only the most important aspects. In addition, there is some information, such as the factors responsible for variable amounts of opiates, that is mentioned twice, that is, it is in two different paragraphs.
Response: As suggested, the introduction section was shortened, and duplicates were removed.
- Lines 119-130: Add some references to justify these statements.
Response: Additional references were included to support lines 119-130.
- Lines 151-156: Everything that has been done in this work is mentioned, but I am missing a paragraph indicating the importance of carrying it out, focusing a little more on why it is going to be done. Emphasize the objective of the work and the main contribution it makes.
Response: Thankful to the reviewer, and, as suggested, the objective of our work has been added to our manuscript and revised accordingly.
Materials and Methods:
- Line 206: I do not quite understand what you mean. For the second and third repetition, the appropriate amount of sample was weighed again, the solvent was added, and it was sonicated or the initially weighed amount was re-extracted a second time with a new volume of solvent?
Response: Revised the sentence to avoid possible confusion. The poppy seeds were extracted with 2.5 mL of solvent, and the supernatant solution was transferred to a 10 mL volumetric flask. Repeated the extraction on the same mark material thrice with 2.5mL each, transferred to the same volumetric flask, and adjusted the final volume to 10mL by addition of blank solvent.
- Section 2.8.2: I do not quite see the difference between the conclusions that would be obtained with the experiments carried out in each of the two paragraphs.
Response: Two experiments were conducted to probe the ideal time and number of wash cycles needed to extract maximum opiates from poppy seeds. In our first study, we have taken 100 µL aliquots from the same sample solution to study the ideal period for maximum extraction efficiency of opiates. Under these settings, we optimized and established ideal solvents, including binary mixtures and temperature, to extract opiates effectively. In contrast, the second study was undertaken to probe the number of wash cycles needed to extract opiates completely from the poppy seeds.
Results and Discussion:
- Section 3.2.: It would be interesting to include a figure such as a bar chart with the amount obtained with each of the solvents used and their corresponding error bars to see if there are significant differences between them and to be able to find the differences between all the possibilities of solvents analyzed.
Response: As suggested by the reviewer, we have included Figure 4 in the revised version of the manuscript to explain the various solvents’ extraction efficiency on opiates.
- Section 3.3.: When developing the chromatographic method, was only the mobile phase optimized? Were different gradients, column temperature, solvent flows also tried to get the most optimal values?
Response: During chromatographic method optimization, we only optimized different gradients (by varying organic solvent proportions) and column temp at ambient and 40 °C.
- Figure 5: The error bars of each seed lot obtained from the two replicates should be included. Moreover, it would be convenient to indicate by means of equal or different letters the existence or not of significant differences between all the different bars of the graph.
Response: As suggested by the reviewer, we have updated Figure 5 (Figure 6 in the revised version) with error bars.
- Lines 398-409: This shows the results obtained descriptively, explaining what is observed in the table. A greater deeper discussion explaining if the results obtained make sense and are consistent, with a greater comparison with the existing literature would be convenient. Have similar concentrations to those reported in the literature been obtained for similar samples?
Response: As suggested by the reviewer, we have discussed the observed results and compared the previously reported poppy samples results (With reference to morphine, codeine, thebaine, papaverine, and noscapine).
- Sections 3.6.1 and 3.6.2.: An inadequate cleaning of the seeds seems to be responsible for the variability in the concentration. However, when washing the seed, the alkaloids to be determined are eliminated. Therefore, how could uniformity in the measurements be obtained so that it was reproducible, and conclusions could be make?
Response: A valuable comment and the authors completely agree with the reviewer. Finely grounded seeds are extracted, and recorded weight of the extractables every time after solvent removal to maintain uniformity of the concentration measurements.
- Lines 602-608: This would fit better at the end of the introduction as the objective and importance of the work to be performed.
Response: The content in lines 602-608 has been moved to the introduction section, as suggested.
Conclusions:
- It is mentioned that only 8 adulterants were analyzed. However, Table 5 lists many more adulterants identified. What is it referring to in each case?
Response: The major emphasis of Table 5 is to showcase the phytochemical composition of eight adulterants of poppy seeds. In most cases, at least two samples of the same plant have been analyzed for reproducibility. For other adulterants, only one sample of Salvia hispanica, Amaranthus paniculatus, and Eschscholzia californica was analyzed for chemical composition and compared with reported literature on these plants.
- In order to study the compounds coming from the adulteration, this adulteration was carried out by the researchers of the work to see the effect that it produces. However, did none of the seeds analyzed show adulteration? Does this adulteration occur after the harvest, before it is marketed, or when?
Response: We agree with the reviewer on the ‘unable to detect’ adulteration with poppy seeds. Even though we did not observe economic adulteration with the current sample set, acquiring and studying every poppy seed in commerce is almost impossible. However, we strongly believe that the results presented in the current manuscript would serve as a proactive analytical strategy to effectively combat the economic adulteration with poppy seeds and be useful in differentiating the natural abundance of opiates from synthetically spiked opiates.
Additionally, you must take care of some formal aspects of the manuscript:
- Lines 165, 183, 201, 222,…: Capitalize each word according to the format of the journal. Unify and apply to the entire manuscript.
- Lines 320 – 328: Review the format of the paragraph, it should not be in bold. Put it in normal and justified font.
- Put a separation after and before “–”, “=”, and “<”. Unify and apply to the entire manuscript.
- Figure 5: The name of the axes is missing.
- Check the references according to the format of the journal. The name of the journal would be always in abbreviated format and italics, and after each abbreviated word there should be a dot “.”.
- References 1, 3, 25, 43, 57, 82: The DOI is missing.
- References 62, 78: The page number is missing.
Response (14-20): As suggested by the reviewer, we have thoroughly checked the manuscript and revised it accordingly.